# The netrin receptor UNC-40/DCC assembles a postsynaptic scaffold and sets the synaptic content of GABA$_A$ receptors

Xin Zhou [1], Marine Gueydan[1], Maelle Jospin[1], Tingting Ji[1,2], Aurore Valfort[1,3], Bérangère Pinan-Lucarré[1,4]✉ & Jean-Louis Bessereau [1,4]✉

Increasing evidence indicates that guidance molecules used during development for cellular and axonal navigation also play roles in synapse maturation and homeostasis. In *C. elegans* the netrin receptor UNC-40/DCC controls the growth of dendritic-like muscle cell extensions towards motoneurons and is required to recruit type A GABA receptors (GABA$_A$Rs) at inhibitory neuromuscular junctions. Here we show that activation of UNC-40 assembles an intracellular synaptic scaffold by physically interacting with FRM-3, a FERM protein ortho-logous to FARP1/2. FRM-3 then recruits LIN-2, the ortholog of CASK, that binds the synaptic adhesion molecule NLG-1/Neuroligin and physically connects GABA$_A$Rs to prepositioned NLG-1 clusters. These processes are orchestrated by the synaptic organizer CePunctin/MADD-4, which controls the localization of GABA$_A$Rs by positioning NLG-1/neuroligin at synapses and regulates the synaptic content of GABA$_A$Rs through the UNC-40-dependent intracellular scaffold. Since DCC is detected at GABA synapses in mammals, DCC might also tune inhibitory neurotransmission in the mammalian brain.

[1] Univ Lyon, Université Claude Bernard Lyon 1, CNRS UMR 5310, INSERM U 1217, Institut NeuroMyoGène, 69008 Lyon, France. [2] Present address: Instrumental analysis center, Shanghai Jiao Tong University, 200240 Shanghai, China. [3] Present address: Center for clinical pharmacology, Saint Louis College of Pharmacy, 2 Pharmacy Place, Saint-Louis MO 63110, USA. [4] These authors contributed equally: Bérangère Pinan-Lucarré, Jean-Louis Bessereau. ✉email: berangere.pinan-lucarre@univ-lyon1.fr; jean-louis.bessereau@univ-lyon1.fr

The precise patterning of synapses arises from a series of spatially and temporally regulated developmental events that include axon navigation, contact recognition of pre- and postsynaptic partners and synaptic differentiation. Axonal navigation is guided by multiple extracellular guidance cues that activate signaling receptors on the surface of the growth cone. Similarly, synapse formation relies on tightly regulated structural and molecular changes that are instructed upon interaction of synaptic adhesion molecules. Intriguingly, numerous evidences also point to a role of axon guidance molecules in synaptic differentiation and plasticity[1–3], reviewed in Poon et al.[4]. Yet, it remains unclear for a given receptor what dictates the switch between a role in neurite guidance and a function in synaptogenesis or synaptic plasticity. Here, we identified the molecular mechanisms implementing the role of the netrin receptor DCC (Deleted in Colorectal Cancer) in the control of synaptic GABA_A receptor content in C. elegans.

Pioneer work in C. elegans initially identified three genes, unc-5, unc-6, and unc-40, that control the directionality of circumferential cell migrations and axon guidance[5] (reviewed by Chisholm et al.[6]). Subsequently, netrins were identified as diffusible chemotropic factors of the laminin superfamily homologous to UNC-6[7,8]. The vertebrate netrin receptors DCC and UNC5 were recognized according to their homology to C. elegans UNC-40 and UNC-5, respectively[9,10]. DCC is a type I transmembrane receptor that belongs to the immunoglobulin superfamily (IgSF). Netrin binding to the DCC ectodomain causes receptor dimerization, which brings in close proximity the cytosolic regions of the receptors and enables their dimerization. This provides a docking platform to recruit activators of multiple signaling pathways, usually leading to an attractive response towards high netrin concentrations (reviewed by Boyer and Gupton[11]). In the presence of UNC5, netrin would trigger the formation of DCC/UNC5 heterodimers that mediate repulsive behaviors[12]. This long-range chemotatic gradient model has recently been revisited after analysis of axonal growth in mice where netrin expression was inactivated in specific subregions of the developing nervous system[13–15]. In these studies, the phenotypes were more consistent with a short-range haptotactic guidance model involving the interaction of growing axons with netrin present in the local environment. Detailed analysis of single axon outgrowth in C. elegans, together with in silico modeling, suggest that UNC-6/netrin biases the distribution of UNC-40/DCC at the membrane surface and stimulates an UNC-40-dependent protrusive activity towards growth direction[16,17].

Scarce reports also demonstrated a role of netrin/DCC signaling at synapses in the mature brain. In cortical pyramidal neurons, DCC is thought to be involved in LTP (long-term potentiation) upon activation of the Src kinase and phosphorylation of the NMDA receptor[1]. Consistently, deletion of DCC in the adult forebrain results in shorter dendritic spines, behavioral defects, and loss of long-term potentiation. Activity-dependent secretion of netrin was recently involved in DCC-dependent potentiation of excitatory glutamatergic transmission in the hippocampus via Ca2+-dependent recruitment of GluA1-containing AMPA receptors[18].

The dual role of DCC in both development and synaptic homeostasis is evolutionarily conserved. In C. elegans, the UNC-6 netrin is secreted by ventral cells, and is thought to form a ventral-to-dorsal gradient[5,19]. C. elegans cells and axons utilize the polarized distribution of UNC-6 to orient circumferential migrations using the UNC-40 and UNC-5 receptors[20]. A well-documented UNC-40-dependent migration process is the outgrowth of body-wall muscle cell expansions towards motoneurons[21]. During post-embryonic development the muscle cells, which are located in four lateral quadrants along the animal's body, extend projections called "muscle arms" to contact motoneurons at the medial ventral and dorsal cords and form en passant neuromuscular junctions (NMJs). UNC-40 drives muscle arm extension in response to the Punctin/MADD-4 guidance cue that is secreted by developing motoneuron axons[22]. On the ventral side, MADD-4 functions redundantly with UNC-6. UNC-40 activation triggers the remodeling of the actin network and involves multiple actors including the Rho guanine-nucleotide exchange factor (GEF) Trio-homolog UNC-73, members of the WAVE actin-polymerization complex and UNC-95, a component of focal adhesion complex[23]. The role of UNC-40/DCC was also carefully analyzed in the migration of several neurons (reviewed by Chisholm et al.[6]).

Besides its canonical functions in guidance, UNC-40 plays direct roles in C. elegans synaptogenesis. Netrin signaling was shown to control synaptic connectivity between the two interneurons AIY and RIA[24]. In this system, UNC-40 guides the migration of postsynaptic neuron RIA and drives presynaptic differentiation in the AIY neuron in response to local secretion of UNC-6 by a glial cell. Downstream signaling modulates actin assembly and recruits presynaptic components[25]. Recently, UNC-6 was involved in the male-specific maintenance of synapses between the sensory neuron PHB and the AVG interneuron[26].

We recently demonstrated that UNC-40 plays a role in the postsynaptic organization of inhibitory NMJs of C. elegans[27]. In C. elegans, each body-wall muscle cell receives excitatory and inhibitory innervation from cholinergic and GABAergic motoneurons. The evolutionarily conserved CePunctin/MADD-4 protein is identified as an anterograde synaptic organizer that specifies the cholinergic versus GABAergic identity of postsynaptic domains[28]. CePunctin belongs to a family of poorly characterized extracellular matrix proteins, the ADAMTS-like proteins that contain multiple thrombospondin type-1 repeat (TSR) and immunoglobulin (Ig) domains as well as structurally unsolved domains in common with the ADAMTS family[29]. The functions of its vertebrate orthologs Punctin1/ADAMTSL1 and Punctin2/ADAMTSL3 are ill-defined, but Punctin2 is expressed in the brain and was identified as a susceptibility gene for schizophrenia[30]. Ce-punctin generates long (L) and short (S) isoforms by use of alternative promoters. MADD-4B/Punctin S and MADD-4L/Punctin L are differentially secreted by GABAergic and cholinergic neurons and trigger postsynaptic clustering of type A GABA receptors (GABA_ARs) and acetylcholine receptors (AChRs), respectively. At the inhibitory NMJ, MADD-4B-dependent clustering of GABA_ARs involves at least two molecular pathways (for review, see Zhou and Bessereau, 2019)[31]. First, MADD-4B binds and clusters the synaptic adhesion molecule NLG-1/neuroligin in front of GABAergic boutons[27,32]. Second, it binds, recruits, and likely activates the netrin receptor UNC-40/DCC, which in turn promotes the interaction of GABA_ARs with neuroligin through a non-characterized mechanism[27]. Since UNC-40 also controls the growth of muscle arms, the inhibitory NMJ represents an interesting paradigm to compare the molecular pathways required for the guidance and the synaptic functions of UNC-40.

Using a genetic strategy, we identify here two proteins, FRM-3 and LIN-2, that implement the function of UNC-40 for the recruitment of GABA_ARs at inhibitory NMJs. We show that UNC-40 recruits FRM-3, a FERM (p4.1, Ezrin, Radixin, Moesin) protein orthologous to FARP1/2, by a physical interaction between the intracellular P3 domain of UNC-40 and the FERM-FA tandem of FRM-3. In turn, FRM-3 recruits LIN-2, the ortholog of CASK (Calcium calmodulin dependent Serine/Threonine kinase), which might provide a hub to physically connect the GABA_ARs, FRM-3/FARP and NLG-1/neuroligin. All these processes are orchestrated by the main synaptic organizer MADD-4B/Punctin S, which both controls the synaptic localization of

GABA$_A$Rs through NLG-1/neuroligin and the synaptic content of GABA$_A$Rs through an UNC-40-dependent intracellular scaffold.

## Results

**Mutations of *frm-3*/FARP and *lin-2*/CASK cause a strong decrease of synaptic GABA$_A$Rs.** To identify synaptic proteins that control GABA$_A$Rs clustering, we mutagenized a *C. elegans* knock-in strain expressing fluorescently tagged GABA$_A$Rs and performed a visual screen for abnormal fluorescence pattern. In this strain, the tagRFP-coding sequence was inserted in the *unc-49* locus, fusing tagRFP to the extracellular N-terminus shared by the three GABA$_A$R subunits UNC-49A, B and C. Among a wide range of phenotypes, we identified two strains with very strong decrease of synaptic fluorescence. Causative mutations were identified by whole genome sequencing, genetic mapping and rescue experiments.

One strain contained a mutation in the *frm-3* gene, which codes for cytosolic proteins orthologous to the mammalian neuronal proteins FARP1 and FARP2 (Fig. 1a and Supplementary Fig. 1a). Two FRM-3 isoforms are generated by alternative splicing that share a common N-terminal part containing a FERM (p4.1, Ezrin, Radixin, Moesin) and FA (FERM-Adjacent) domains. FRM-3A does not contain any recognizable domain in its C-terminal part. The *frm-3B* transcript was more recently annotated in WormBase and not characterized to our knowledge. FRM-3B contains a GEF domain and two PH domains as in mammalian FARPs. The *kr319* mutation isolated in our screen introduces a G to A point mutation that inactivates a splicing donor site. For subsequent characterization of *frm-3*, we used the null allele *frm-3(gk585)*, further referred to as *frm-3(0)*, which deletes most of the FERM coding region and introduces an early STOP (Fig. 1a). The second strain that was analyzed contained a mutation in the *lin-2* gene, which encodes a membrane-associated guanylate kinase (MAGUK) orthologous to CASK. The *lin-2* locus generates two isoforms by the use of two different promoters. LIN-2A contains a CaM-kinase domain, two LIN-27 domains, a PDZ domain, a SH3 domain and a guanylate kinase domain. LIN-2B lacks the N-terminal CaM-kinase domain, and one of the LIN-27 domain (Fig. 1b). The *lin-2(kr357)* retrieved in our screen contained a deletion spanning the 1218-1331 nt, which caused a frameshift from valine 122 with subsequent introduction of an early stop codon. Interestingly, *frm-3* and *lin-2* were formerly shown to impact UNC-49 content at synapses, but the phenotypes that we observed seemed much more dramatic than previously reported[33].

To confirm our observations, we analyzed GABA$_A$Rs by immunostaining[34] and detected a strong loss of UNC-49 staining at synapses in both *frm-3(0)* and *lin-2(n397)* mutants as compared with wild-type (WT) animals (Fig. 1c). We then used the *RFP::unc-49* allele to perform a quantitative analysis of synaptic receptor content. In both mutants, fluorescence was decreased by about 80 % in synaptic regions as compared with WT. GABA$_A$Rs were barely detectable, forming extremely small puncta at GABAergic boutons (Figs. 1d–e and Supplementary Fig. 1d). By contrast, the number and size of presynaptic GABA boutons were unaltered based on the quantitative analysis of the synaptic marker SNB-1-GFP specifically expressed in GABAergic motoneurons (Fig. 1d and Supplementary Fig. 1b–c).

In *C. elegans*, muscle cells send projections to the motoneurons, named muscle arms, and establish *en passant* synapses at the dorsal and ventral nerve cords. To exclude that GABA$_A$R decrease might be explained by muscle arm development defects, we analyzed muscle morphology in the *frm-3(0)* mutant and found that the number and the morphology of muscle arms were

normal (Supplementary Fig. 1e–f). The *frm-3* mutation could hamper the biosynthesis or stability of GABA$_A$Rs. We therefore measured the protein level of RFP-tagged UNC-49 by western blot and we found that the amount of UNC-49 was normal in *frm-3(0)* mutant (Supplementary Fig. 1g-h). These results are consistent with previously published electrophysiological recordings showing that bath application of GABA elicits similar currents in wild-type and *frm-3(0)* mutants[33]. To evaluate the functional consequence of *frm-3* and *lin-2* inactivation on GABAergic transmission, we recorded GABA-evoked currents in muscle upon optogenetic stimulation of GABA motoneurons. As compared with the WT, we observed a 75% and 90% decrease of GABA-evoked currents in *frm-3* and *lin-2* null mutants, respectively (Fig. 1f–g). Altogether, these data indicate that FRM-3 and LIN-2 are essential for the localization of GABA$_A$Rs at inhibitory NMJs.

To further characterize the contribution of *lin-2* to GABA$_A$R clustering, we analyzed several *lin-2* alleles. In *lin-2(n397)*, where amino acids 137-463 of LIN-2A as well as part of LIN-2B N-terminus were deleted, the synaptic fluorescence level of RFP-UNC-49 was reduced by 75%, similar to *frm-3(0)* mutants (Fig. 1h). Similarly, in *lin-2(n105)*, where the last 22 amino acids of the C-terminus common to LIN-2A and LIN-2B were deleted, GABA$_A$Rs were also decreased by 70% at either 20 °C or 25 °C (Fig. 1h), despite the reported thermosensitive vulva-less phenotype of these mutants. Yet the *lin-2(e1309)* mutant, which has been used as a reference allele in previous studies[33], only caused a 40% reduction in RFP-UNC-49 fluorescence. Partial information on the *lin-2(e1309)* allele suggests that LIN-2A coding sequence is altered but that LIN-2B might still be expressed, since LIN-2B coding sequence and part of the LIN-2B promoter region might be intact (Fig. 1b)[35,36]. To further characterize the contribution of individual *lin-2* isoforms to GABA$_A$R clustering, we expressed LIN-2A or LIN-2B in the muscle cells of *lin-2(n397)* mutants and monitored RFP-UNC-49 fluorescence levels. Expressing either LIN-2A or LIN-2B completely rescued the *lin-2(n397)* mutant phenotype, indicating that the domains within the short LIN-2B isoform are sufficient for GABA$_A$R clustering (Fig. 1h). Altogether, these data indicate that LIN-2 acts cell autonomously to control the clustering of GABA$_A$Rs.

**FRM-3 and LIN-2 localize at synapses independently from neuroligin.** To analyze the subcellular distribution of FRM-3 and LIN-2, we built a series of single-copy transgenes to express fluorescently tagged isoforms of FRM-3 and LIN-2 in muscle. FRM-3A and LIN-2A muscle-specific reporters formed puncta along the nerve cords (Fig. 2a–c). FRM-3A-GFP and RFP-LIN-2A highly colocalized (Fig. 2d). FRM-3 and LIN-2 puncta localized at GABAergic synapses (Fig. 2a), but were also found in between GABAergic synapses (Fig. 2a, d). Consistently, FRM-3B-GFP and GFP-LIN-2A were also present at cholinergic NMJs (Supplementary Fig. 2a–c).

We previously showed that NLG-1/neuroligin controls the synaptic targeting of GABA$_A$R[27]. We confirmed this observation using a novel *unc-49::pHluorin* knock-in strain, which visualizes only GABA$_A$Rs at the plasma membrane. In *nlg-1(0)* null mutant animals, the overall fluorescence level of pHluorin-UNC-49 was decreased by 50% and small GABA$_A$R clusters were fragmented and diffused away from GABA synapses (Fig. 2e–g). We then tested whether NLG-1 controls FRM-3 and LIN-2 positioning at GABAergic synapses. In *nlg-1(0)* mutants, FRM-3B-GFP, and GFP-LIN-2A were present at excitatory and inhibitory synapses. The fluorescence level of FRM-3B-GFP was unchanged in *nlg-1(0)* mutants while the RFP-LIN-2 fluorescence level was slightly

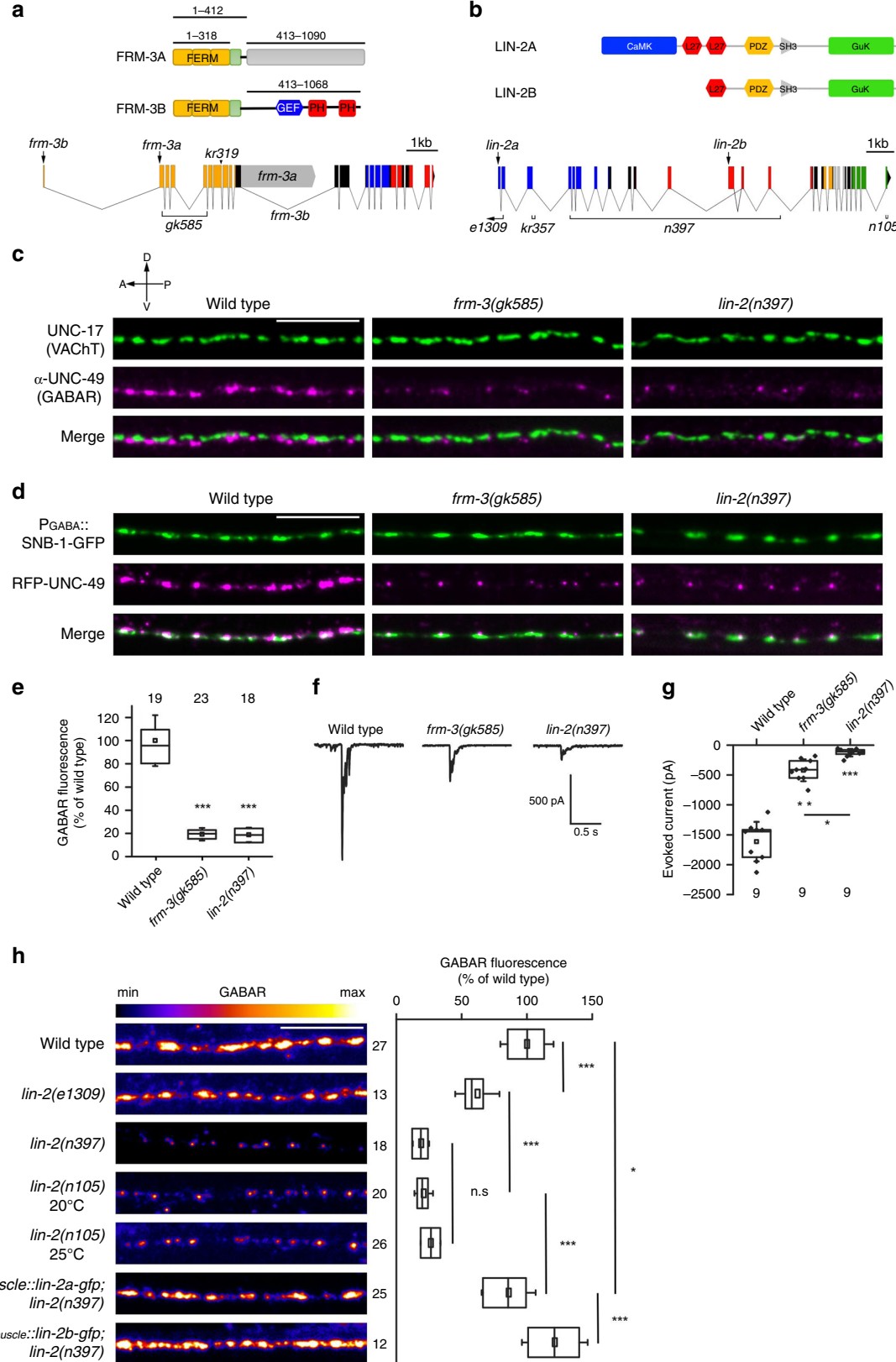

reduced (Fig. 2h–k). Conversely, NLG-1-GFP fluorescence level was nearly WT in *lin-2(0)* or *frm-3(0)* mutants (Fig. 2l). Altogether these data suggest that NLG-1 is not required for synaptic targeting of FRM-3 but might participate to the localization of LIN-2 at GABAergic synapses.

**Postsynaptic UNC-40/DCC recruits FRM-3 independently from LIN-2.** We previously showed that UNC-40/DCC signaling promotes the recruitment of GABA$_A$Rs onto NLG-1 clusters[27]. Using *rfp-* and *pHluorin-unc-49* knock-in stains, we confirmed that the amount of either total or surface GABA$_A$Rs was reduced

**Fig. 1 Mutations of *frm*-3 and *lin*-2 cause a strong loss of synaptic GABA$_A$Rs. a** Predicted domains of FRM-3A and FRM-3B: FERM (4.1, Ezrin, Radixin, Moesin, yellow box), FA (FERM Adjacent, green box), Guanine Nucleotide Exchange Factor (blue), Pleckstrin homology (red), unknown domain (gray). Domain boundaries are indicated on the top (amino acid position). Bottom: structure of the *frm-3* locus. Molecular information for *frm-3* and *lin-2* mutants are available in the Methods. **b** Predicted domains of LIN-2A and LIN-2B: CaMK (CaM-kinase), L27 (LIN-27), PDZ, SH3 (SRC Homology 3), GuK (guanylate kinase) domains. Bottom: structure of the *lin-2* locus. **c** Immunostaining of cholinergic boutons (anti-vesicular acetylcholine transporter (VAChT)/UNC-17 antibody) and GABA$_A$Rs (anti-UNC-49 antibody) in wild-type, *frm-3(gk585)* and *lin-2(n397)* animals. **d** Confocal imaging of strains expressing the synaptic vesicle marker SNB-1-GFP in GABAergic boutons and RFP-labeled GABA$_A$Rs (*unc-49::rfp* knock-in) in wild-type, *frm-3(gk585)* and *lin-2(n397)* backgrounds. **e** Fluorescence intensity of RFP-UNC-49 in wild-type, *frm-3(gk585)* and *lin-2(n397)* animals. Data are presented as box plots showing lower and upper quartiles (box), mean (square), median (center line) and standard deviation (whiskers). Data are normalized to the wild type. One-way ANOVA followed by Tukey's multiple comparison tests of each group as compared with the wild type. ***$p < 0.001$. **f, g** Representative traces and values of currents evoked by 10 ms light stimulation recorded on muscle cells of wild-type, *frm-3(gk585)* and *lin-2(n397)* worms expressing ChR2 in GABA motoneurons. **g** Box plot as in Fig. 1e; individual values are shown (diamonds). Kruskal Wallis test, $p < 0.0001$, followed by Dunn's post-tests: wild type/*frm-3* **$p < 0.01$, wild type/*lin-2* ***$p < 0.0001$, *frm-3*/*lin-2* *$p < 0.05$. **h** RFP-GABA$_A$R expression in various *lin-2* mutants. GFP-LIN-2A or GFP-LIN-2B fusions were expressed under the muscle-specific promoter *Pmyo-3*. GABA$_A$R fluorescence levels were normalized to the wild type. Box plot as in Fig. 1e. One-way ANOVA followed by Turkey's multiple comparison tests. *$p < 0.05$, ***$p < 0.001$, n.s: not significant. In this figure and all other figures, pictures are sums of Z-stacks acquired at a spinning disk confocal microscope; in all figures, anterior is to the left and dorsal is up, scale bars = 10 μm. The numbers of animals are indicated on the box plots.

in multiple loss-of-function *unc-40* allele mutants, including the reference allele *e1430* further referred to as *unc-40(0)* (Fig. 3a–b and Supplementary Fig. 3a–c). We hypothesized that FRM-3 and LIN-2 might be involved in this mechanism. We first tested if UNC-40/DCC could regulate the synaptic content of FRM-3 and LIN-2. In *unc-40(0)* null mutants, RFP-LIN-2A and FRM-3B-GFP were reduced by approximately 50% and 70%, respectively (Fig. 3a–d), whereas their localization was not changed (Fig. 3e–f). This decrease was not a consequence of GABA$_A$R decrease since FRM-3B-GFP and GFP-LIN-2 were properly expressed in *unc-49(0)* mutants (Supplementary Fig. 4a–e).

FRM-3 and LIN-2 might be both independently regulated by UNC-40, or one of these two components might be regulated by UNC-40, and in turn recruit the second one. To distinguish these two hypotheses, we investigated how LIN-2 and FRM-3 regulate each other. We found a strong loss of LIN-2 in *frm-3(0)* mutants (Fig. 3g–i). By contrast, the FRM-3B isoform, or the FERM-FA domains only, formed proper synaptic puncta in *lin-2(n397)* mutant animals (Fig. 3h, j–k). These observations suggest that UNC-40 controls the synaptic localization of FRM-3, which in turn recruits LIN-2.

We then tested the hierarchy between UNC-40 and FRM-3. Both proteins colocalized at NMJs (Fig. 3l). However, although the removal of UNC-40 reduced FRM-3, the loss of FRM-3 did not affect UNC-40 localization or content (Fig. 3m–n). These data suggested that UNC-40 primarily localizes at synapses and subsequently recruits FRM-3 at synapses. To strengthen this hypothesis, we overexpressed in muscle cells a myristoylated fusion of the intracellular domain of UNC-40, which acts as constitutively active receptor. Myr-UNC-40 distributes at the entire surface of the cell and causes the budding of exuberant membrane processes[27,37]. In *unc-40(0)* mutants, FRM-3B-GFP was weakly detected in non-synaptic regions. Co-expression of myr-UNC-40 caused strong accumulation of FRM-3 at the membrane in the budding regions where myr-UNC-40 accumulated (Fig. 3o, p). Altogether, these data suggest that activation of UNC-40 is sufficient to recruit FRM-3.

**FERM-FA domain of FRM-3 is required in the muscle cell to control GABA$_A$R clustering.** We investigated the function of the different FRM-3 domains for GABA$_A$R clustering. Muscle-specific expression of either FRM-3A or FRM-3B in *frm-3(0)* mutants rescued the synaptic content of GABA$_A$Rs (Fig. 4a). Both isoforms encompass the same N-terminal moiety that contains the predicted FERM and FA domains. Expression of these domains was sufficient to rescue the loss of GABA$_A$Rs in *frm-3(0)*

mutants, while the FERM domain only or the C-terminus of FRM-3B containing the RhoGEF and the two PH domains failed to rescue (Fig. 4a). Consistently, FRM-3A, FRM-3B, and FERM-FA domains were properly targeted to synapses while the FERM domain alone and the C-terminus of FRM-3B were diffusely localized in the muscle near the nerve cord (Fig. 4b). The C-terminus of FRM-3A was present in the nucleus of muscle cells, probably through unmasking of a cryptic nuclear localization signal (Supplementary Fig. 5a–b). Altogether, these data indicate that the FERM-FA domains of FRM-3 are necessary and sufficient to promote GABA$_A$R clustering.

A recent study reported that the FERM-FA domains of the *Drosophila* protein Yurt form oligomers, and that multimerization of Yurt supports its function in cell polarity[38]. Oligomerization relies on several hydrophobic residues within the F3 lobe of the FERM domain, which are conserved in FRM-3 (Supplementary Fig. 1a). We investigated whether the FERM-FA domains of FRM-3 might oligomerize using GST-pull down assays. GST-tagged FERM-FA domains efficiently pulled down HA-tagged FERM-FA (Fig. 4c). As a positive control, we verified that GST-tagged LIN-2 pulled down the HA-tagged FERM-FA domains (Fig. 4c), consistently with a previously reported yeast-2-hybrid interaction between LIN-2 and FRM-3[33]. Therefore, the FERM-FA domains of FRM-3 can oligomerize, which might in turn assemble a submembrane lattice promoting GABA$_A$R clustering.

**UNC-40 physically interacts with FRM-3 through its P3 domain.** Our data show that activation of UNC-40 can recruit FRM-3 to the plasma membrane. We tested if these two proteins form a complex, first by co-immunoprecipitation experiments using strains expressing UNC-40 and different FRM-3 versions. UNC-40 co-immunoprecipitated with FRM-3B full length or the FERM-FA domains (Fig. 4d–e). Second, we used in vitro GST pulldown assay and detected the direct binding of FERM-FA to the UNC-40 intracellular domain (Fig. 4g). Interestingly, the FERM domain of myosin-X binds the mammalian DCC through its C-terminal P3 domain[39,40]. This domain folds into an α-helix which contains hydrophobic residues critical for binding to the FERM domain of myosin X[39,40]. Despite poor conservation of the primary sequence, this α-helix can be readily predicted in UNC-40 with a remarkable conservation of hydrophobic residues between UNC-40 and DCC (Fig. 4f)[37]. To test if P3 is involved in the UNC-40/FRM-3 interaction, we repeated the in vitro GST pulldown after deleting P3 from the UNC-40 intracellular domain and could no longer detect an interaction with the FERM-FA domain of FRM-3 (Fig. 4g).

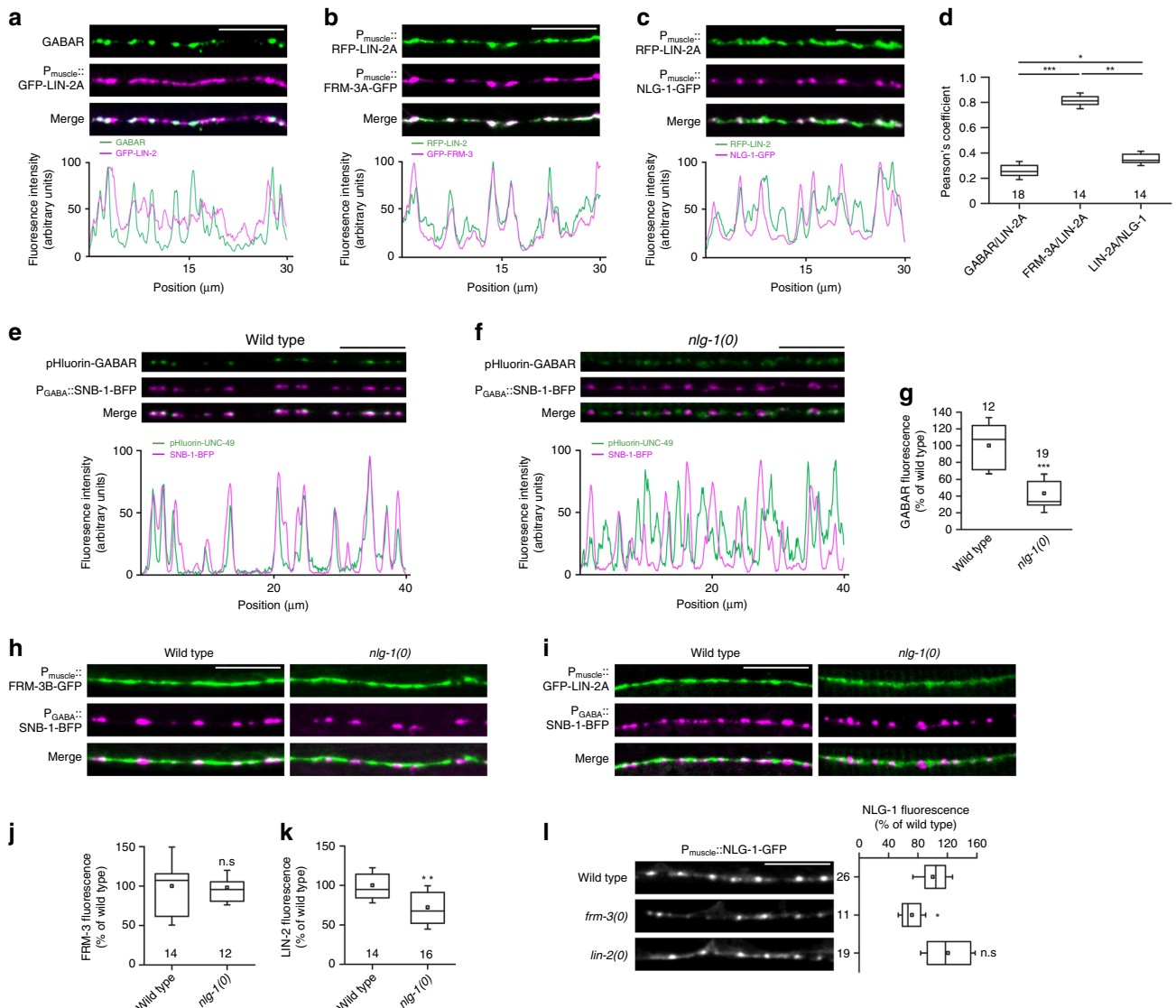

**Fig. 2 LIN-2 and FRM-3 colocalize at neuromuscular junctions. a** Confocal detection of RFP-labeled GABA$_A$R expressed from the *unc-49::rfp* knock-in allele *kr296* and of GFP-LIN-2A expressed under the control of the muscle-specific promoter *Pmyo-3*. Fluorescence profiles indicate GFP-LIN-2A and RFP-UNC-49 fluorescence intensities along the nerve cord from the pictures above. **b** Confocal detection of RFP-LIN-2A and FRM-3A-GFP specifically expressed in muscle. The fluorescence profiles indicate RFP-LIN-2A and FRM-3A-GFP fluorescence intensities along the nerve cord from the pictures above. **c** In muscle cells, RFP-LIN-2A is present at GABAergic synapses labeled with NLG-1-GFP. **d** Pearson's correlation coefficient between different markers. Box plots show lower and upper quartiles (box), median (center line) and standard deviation (whiskers). Kruskal–Wallis test ($p < 0.001$) followed by Dunn's post test. ***$p < 0.001$; **$p < 0.01$; *$p < 0.05$. **e–g** Detection of surface-expressed GABA$_A$R in the *unc-49(kr378::pHluorin)* knock-in strain shows small and diffuse clusters in the *nlg-1(0)* null mutant (**f**) as compared with the wild type (**e**); SNB-1-BFP labels GABAergic boutons (**g**) pHluorin-UNC-49 fluorescence levels were quantified and normalized to the wild type. Box plot as in Fig. 1e. Two-tailed Student's *t* test ***$p < 0.001$. **h–k** Confocal detection of FRM-3B-GFP and GFP-LIN-2A expressed in muscle in wild-type animals or in *nlg-1(ok259)* mutants; SNB-1-BFP labels GABAergic boutons. **j, k** Quantification of FRM-3-GFP and GFP-LIN-2A fluorescence levels. Box plots are as in Fig. 1e. Two-tailed Student's *t* test **$p < 0.01$. n.s: not significant. **l** Confocal detection of NLG-1-GFP expressed in the muscle of wild-type, *frm-3 (gk585)* and *lin-2(n397)* animals. The fluorescence level of NLG-1-GFP was quantified and normalized to the wild type. Box plot as in Fig. 1e. One-way ANOVA followed by Tukey's multiple comparison tests of each group compared with the wild type. *$p < 0.05$, n.s: not significant. Scale bars = 10 µm.

The P3 domain of DCC is necessary for intracellular multimerization in vitro and for axonal attraction in *Xenopus* spinal neurons[41]. To test a putative role of P3 at GABA synapses, we replaced the P3 domain of UNC-40 by an HA-tag using CRISPR/Cas9 (Fig. 5a). As compared with *unc-40(e1430)* null mutants that are small and have severe locomotion and egg-laying defects, *unc-40::ΔP3* mutants had normal body length and exhibited much less defects in locomotion and egg-laying (Fig. 5b). To quantitively measure the remaining activity of UNC-40ΔP3, we scored post-embryonic muscle arm growth and ventral guidance of the mechanosensory AVM axon, which were shown to rely mostly on

UNC-40[20,23]. The *unc-40::ΔP3* mutants had on average 2 muscle arms per muscle cells, as compared with only 1 in *unc-40* null mutants and 4 in the wild type (Fig. 5c, d). AVM defects were detected in 15 % of the *ΔP3* animals, as compared with 36 % in null mutants (Fig. 5e, f). Hence, deletion of the P3 domain only partially impairs the function of UNC-40 for attraction guidance.

During axonal guidance UNC-40 controls F-actin dynamics through downstream effectors[25]. We therefore analyzed actin distribution in muscle cells of WT and *unc-40* mutants using the BFP-tagged actin binding protein COR-1/Coronin[42,43]. Actin formed readily distinguishable bundles in muscle arms and along

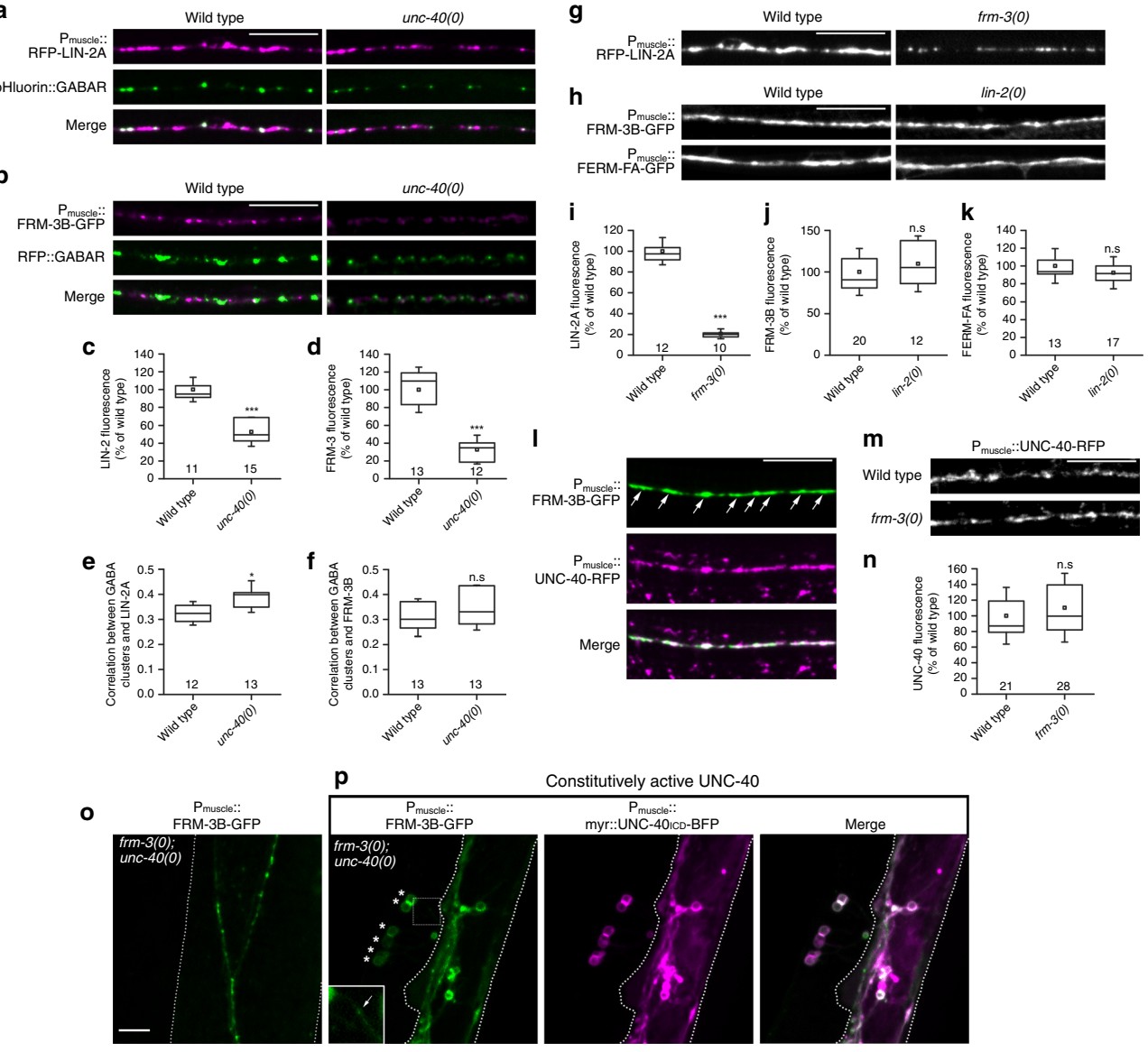

**Fig. 3 The UNC-40/DCC receptor recruits FRM-3 at synapses. a**, **b** Confocal detection of RFP-LIN-2A (**a**) and FRM-3B-GFP (**b**) expression in muscle cells of wild-type and *unc-40(e1430)* mutants; GABA$_A$Rs labeled with pHluorin (**a**) or RFP (**b**) were expressed from the *unc-49(kr397::pHluorin)* and *unc-49 (kr296::RFP)* knock in alleles, respectively. **c**, **d** Quantification of RFP-LIN-2A and FRM-3B-GFP fluorescence levels normalized to the wild type. Box plot as in Fig. 1e. Two-tailed Student's *t* test ***$p < 0.001$. **e**, **f** Pearson's correlation coefficient between GABA$_A$R clusters and LIN-2A or FRM-3B in the wild type and *unc-40(0)* mutants. Box plot as in Fig. 2d. Mann–Whitney test. *$p < 0.05$, n.s: not significant. **g** Muscle-expressed RFP-LIN-2 forms very weak puncta in *frm-3(0)* null mutants as compared with the wild type. **h** Detection of FRM-3B-GFP and FERM-FA-GFP expressed in muscle cells of wild-type and *lin-2(n397)* animals. **i–k** Quantification of RFP-LIN-2A, FRM-3B-GFP and FERM-FA-GFP fluorescence levels normalized to the wild type. Box plots are as in Fig. 1e. Two-tailed Student's *t* test. ***$p < 0.001$. n.s: not significant. **l** Confocal detection of UNC-40-RFP and FRM-3B-GFP expressed in muscle. Arrows point to neuromuscular junctions. **m** Confocal detection of UNC-40-RFP expressed in the muscle of wild-type and *frm-3(gk585)* animals. **n** Quantification of UNC-40-RFP fluorescence levels normalized to the wild type. Box plot as in Fig. 1e. Two-tailed Student's test. n.s: not significant. **o** FRM-3B-GFP expressed in the muscle cell of *frm-3(gk585); unc-40(e1430)* double mutants is detected at cell boundaries. **p** Muscle-specific expression of the myristoylated intracellular domain of UNC-40 (myr::unc-40ICD-BFP), a constitutively active form of UNC-40, causes abnormal blebbing of muscle cells (stars), with thin connections to the muscle-cell body (arrow). myr::unc-40ICD-BFP accumulates along the muscle cell membrane and at blebbing sites, where it attracts FRM-3B-GFP. Dotted line: border of muscle cells. Scale bars = 10 μm.

the nerve cords in WT. This network was present in *frm-3(0)* mutants while it completely disappeared in *unc-40(0)* mutants (Fig. 5g). In *unc-40::ΔP3* animals, actin labeling was decreased but remained detectable in muscle arms and at the proximal part of muscle arm extension at the cord. These data pointed to a remaining activity of UNC-40ΔP3 for controlling actin cytoskeleton. By contrast, the loss of GABA$_A$Rs was indistinguishable in

*unc-40::ΔP3* and in *unc-40* null mutants (Fig. 5h–i). Similarly, the residual amount of FRM-3 FERM-FA was the same at the cords of *unc-40::ΔP3* and *unc-40* null mutants (Fig. 5h, j).

Altogether, these data suggest that UNC-40 can recruit FRM-3 at synapses through physical interaction between P3 and FERM-FA, which represents a limiting step to further promote the clustering of GABA$_A$Rs at synaptic sites.

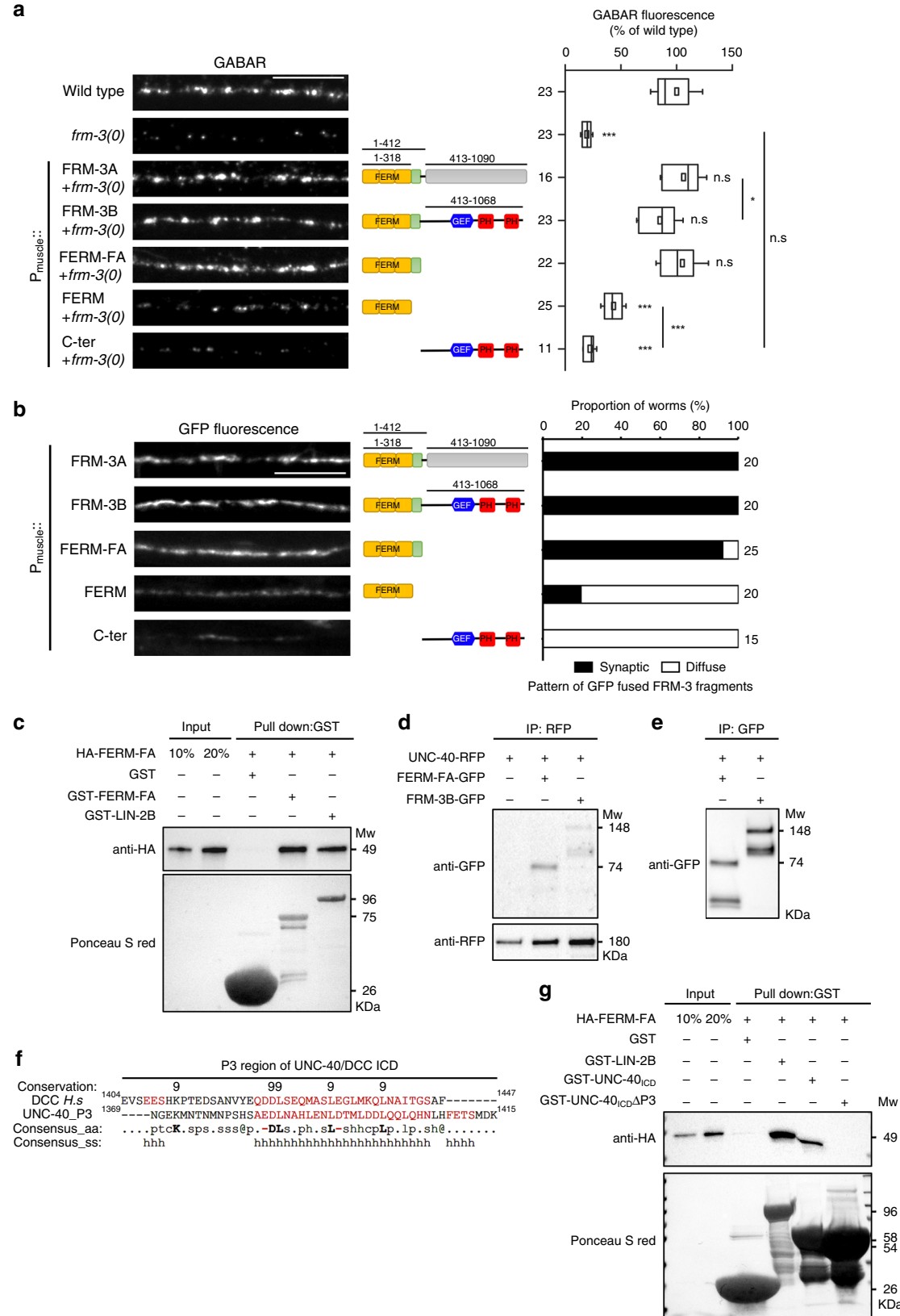

**LIN-2/CASK bridges GABA$_A$Rs with FRM-3 and NLG-1.** Although UNC-40 and FRM-3 are necessary to recruit LIN-2 and GABA$_A$Rs at the nerve cords, we and others[27,32] demonstrated that the PDZ-binding motif of the neuroligin NLG-1 was critical for the synaptic localization of GABA$_A$R. We confirmed these data with our *unc-49::rfp* knock-in strain (Fig. 6c–d). Unlike mammalian neuroligins that end with a type I PDZ motif not predicted to interact with CASK, NLG-1 ends with a type II

**Fig. 4 The FERM-FA tandem of FRM-3 acts cell-autonomously to support GABAAR clustering. a** Confocal detection of RFP-labeled GABAAR expressed from the *unc-49::rfp* knock-in locus *kr296* in *frm-3(0)* mutants alone and upon muscle-specific expression of full-length and truncated versions of FRM-3-GFP (domain schematic as in Fig. 1a). Fluorescence intensity of RFP-UNC-49 was normalized to the wild type. Box plot as in Fig. 1e. One-way ANOVA followed by Tukey's multiple comparison tests of each group compared with the wild type. ***$p < 0.001$; *$p < 0.05$; n.s: not significant. **b** Analysis of FRM-3-GFP distribution in *frm-3(0)* mutants expressing full-length and truncated versions of FRM-3-GFP. The proportion of animals with synaptic or diffuse GFP distribution was scored for each chimera. **c** GST pull-down analysis of FERM-FA domain dimerization. GST-LIN-2B was used as a positive control. The samples were analyzed by immunoblotting using an anti-HA antibody. The same membrane was stained with Ponceau S red to show GST expression. Molecular weights (Mw) are shown on the right. **d, e** In vivo co-immunoprecipitation experiments between UNC-40 and FRM-3 full length or FERM-FA domain. UNC-40-RFP was precipitated from whole worm extracts by RFP-trap-A bead. GFP fused components were detected by western blot. Precipitation efficiency was tested by Western Blot using an anti-RFP antibody. Specificity of co-IP detection was shown in **e**. **f** Alignment and predicted secondary structure of human DCC and UNC-40 P3 region, based on PROMALS3D analysis. Consensus_ss (secondary structure): "h" shows residues involved in an alpha-helix structure; the corresponding amino acid are marked in red. Consensus_aa: aliphatic: l; aromatic: @; hydrophobic: h; polar residues: p; tiny: t; small: s; negatively charged: -; charged: c. **g** GST pull-down analysis of the FERM-FA domain interaction with UNC-40ICD and UNC-40ICDΔP3. GST fused to LIN-2B was used as positive control. Samples were analyzed by immunoblotting using an anti-HA antibody. The same membrane was stained by Ponceau S red to shown GST expression. Molecular weights (Mw) of corresponding proteins are shown on the right. Scale bars = 10 μm.

PDZ-binding domain (Fig. 6a). Because LIN-2 contains a type II PDZ domain, we asked whether these two proteins could physically interact. Using a yeast-2-hybrid assay we detected an interaction between the LIN-2B full length isoform and the intracellular part of NLG-1 requiring the PDZ-binding motif of NLG-1 (Fig. 6a–b). This suggests that a direct connection of NLG-1 and LIN-2 is involved for efficient GABAAR clustering.

How is UNC-49 recruited to the NLG-1-containing complex? Previous work suggested the existence of a direct interaction between FRM-3 and the intracellular TM3-TM4 loop of UNC-49B[33]. Yet, FRM-3 is also required for the recruitment of LIN-2 at synapses. Consistently, we detected strong interaction between the FERM-FA domain of FRM-3 and LIN-2B in a GST pull-down assay (Fig. 4c), in agreement with previous yeast-2-hybrid data[33]. However, in *lin-2(0)* mutants, FRM-3 is still detected at synapses while UNC-49 is decreased to the same extent as in *frm-3(0)* mutants, suggesting that FRM-3 is not sufficient to recruit UNC-49 receptors. Since LIN-2 is also decreased in *frm-3(0)* mutants, we wondered if LIN-2 and UNC-49 might directly interact. First, we performed a GST pull-down assay and demonstrated that HA-tagged LIN-2B interacts with the TM3-TM4 intracellular loop of UNC-49 fused to GST (Fig. 6e). This interaction remained after switching the GST and HA epitopes in a complementary pull-down experiment (Fig. 6f). We then mapped the LIN-2B domains involved in interacting with UNC-49. Deletion of the SH3 domain completely disrupted the interaction with the TM3-TM4 of UNC-49. Deletion of the Guanylate Kinase domain partially impaired the interaction while deletion of the PDZ domain had no effect. Altogether these data indicate that LIN-2 might act as a scaffolding molecule that physically bridges NLG-1 and the GABAAR through FRM-3-dependent cytoplasmic interactions.

This model has at least three predictions. First, the interaction of NLG-1 and UNC-49 should depend on UNC-40 because UNC-40 recruits FRM-3 and LIN-2 at synapses. We thus compared the efficiency of retrieving RFP-UNC-49 after immuno-precipitation of NLG-1-GFP expressed either in wild-type or *unc-40* mutants. We observed that *unc-40* disruption indeed impaired the interaction between UNC-49 and NLG-1 (Fig. 6g, i). Second, the interaction between NLG-1 and UNC-49 should depend on FRM-3 and LIN-2 if they provide a hub to link NLG-1 and UNC-49. Accordingly, the coIP efficiency between NLG-1 and UNC-49 was reduced in *frm-3(0)* or *lin-2(0)* mutant animals (Fig. 6h, j). Third, the model predicts that mutations in *lin-2* and *frm-3* should be epistatic to *nlg-1* mutations. Accordingly, we observed a more severe reduction of RFP-UNC-49 in *nlg-1(0); lin-2(0)* or *frm-3(0)* double mutants than in *nlg-1(0)* single mutants, similar to *lin-2* or *frm-3* single mutants (Supplementary Fig. 6). Interestingly, the *lin-2(0); frm-3(0)* double mutant is also comparable to *lin-2* or *frm-3*

single mutants, suggesting that they indeed function in the same pathway.

**MADD-4/Punctin controls the synaptic localization of the FRM-3-LIN-2 complex.** Previous work in our laboratory identified MADD-4/Punctin as a master organizer of NMJs[28]. The MADD-4L (long) and MADD-4B (short) isoforms govern specific clustering of GABA and AChRs at inhibitory and excitatory NMJs. At GABA synapses, MADD-4B recruits NLG-1 on one hand and activates UNC-40 signaling on the other hand[27]. Whether MADD-4 would control the formation of the post-synaptic FRM-3-LIN-2 scaffold became an appealing hypothesis.

To address this question, we analyzed the synaptic localization of FRM-3 and LIN-2 in *madd-4(0)* (null), *madd-4B(0)* (a mutant in which the *madd-4B* isoform is specifically disrupted but *madd-4L* is still expressed) or *madd-4L(0)* (specific mutation of the *madd-4L* isoform) mutant animals. In *madd-4(0)* the synaptic localization of FRM-3 and LIN-2 was severely compromised, with a fluorescence level decreased by approximately 75 % (Fig. 7a–b, g, Supplementary Fig. 7a). We previously showed that specific disruption of MADD-4B, the only MADD-4 isoform present at GABA synapses, relocalizes GABAARs and NLG-1/neuroligin from GABAergic to cholinergic synapses[27]. In *madd-4B(0)* animals, FRM-3A-GFP and GFP-LIN-2A fluorescence levels were normal but FRM-3 and LIN-2 puncta were relocalized outside of GABAergic synaptic regions (Fig. 7c–d, Supplementary Fig. 7b–c). By contrast, the *madd-4L(0)* mutation did not modify the level and localization of FRM-3 or LIN-2 (Fig. 7e–h, Supplementary Fig. 7d). Altogether these results indicate that MADD-4 controls the formation of the LIN-2-FRM-3 scaffolding complex at inhibitory and excitatory NMJs and that MADD-4B specifies its localization at GABAergic synapses.

**Discussion**

Here we show that UNC-40/DCC plays a pivotal role at GABA synapses by transducing the effect of the extracellular matrix protein Punctin/MADD-4. Upon activation, UNC-40 triggers the formation of an intracellular scaffold, which in turn promotes the interaction of GABAARs with prepositioned Neuroligin/NLG-1 synaptic clusters (graphical model provided in Fig. 8). In this system, Punctin plays a dual role: (i) it specifies the positioning of post-synaptic inhibitory domains in register with GABA release sites by interacting with neuroligin and inducing its clustering in the postsynaptic membrane; (ii) it concentrates UNC-40/DCC at the tip of the muscle arms in synaptic regions, which triggers membrane recruitment of the cytosolic adaptor FRM-3/FARP, likely through direct interaction between the P3 domain of UNC-40 and the FERM domain of

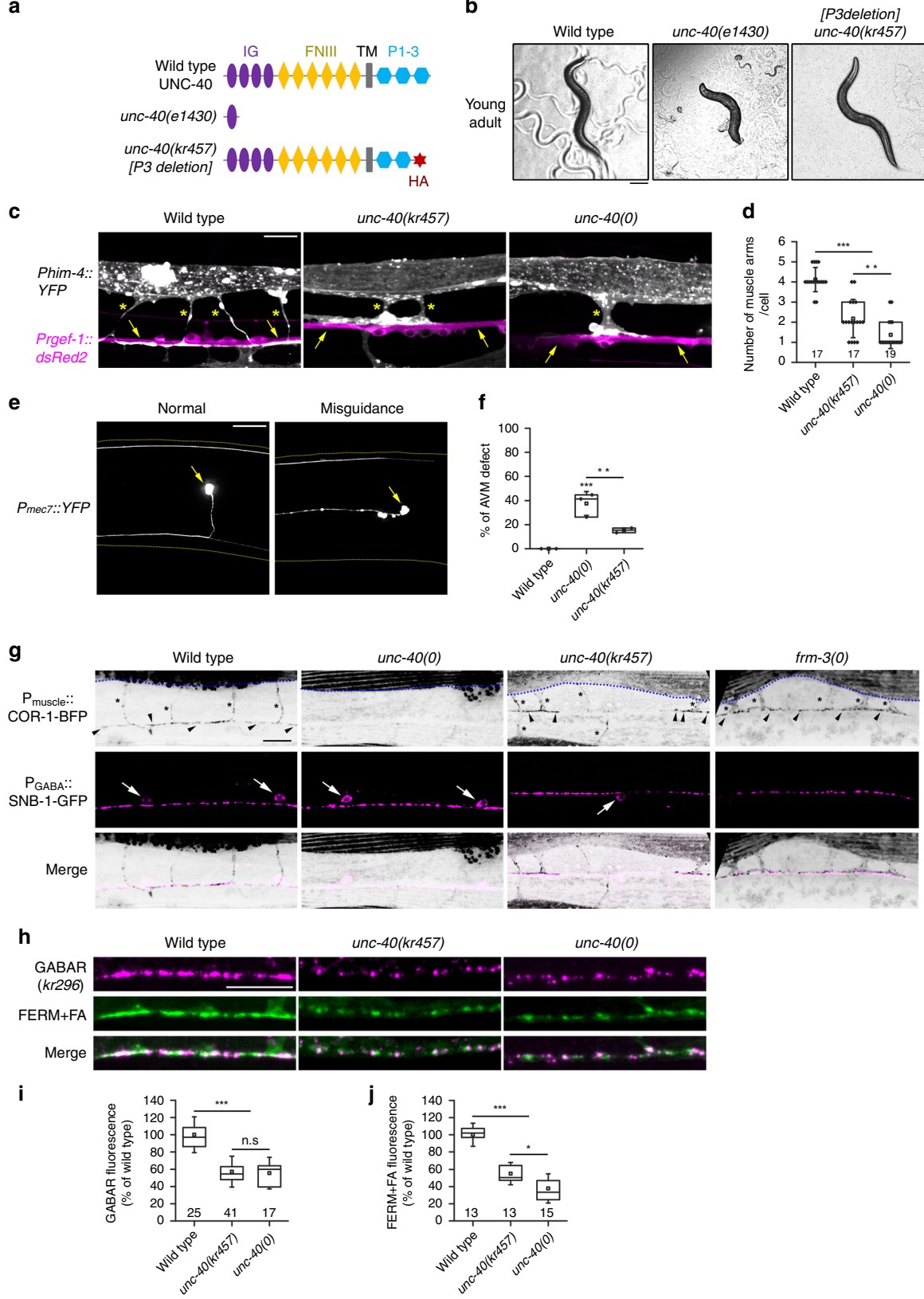

FRM-3. This in turn recruits the scaffolding molecule LIN-2/CASK, which can interact with the UNC-49 GABA$_A$R and the cytoplasmic tail of neuroligin. As a net effect, UNC-40/DCC controls the synaptic content of receptors at GABA synapses in response to Punctin.

Only few studies, so far, have analyzed the role of DCC in synaptic organization. Netrin was shown to have a synaptogenic activity in the mammalian forebrain by stimulating dendrite and axon outgrowth, promoting synaptic adhesion and reorganizing the actin cytoskeleton to foster the clustering of pre-and

**Fig. 5 The P3 domain of UNC-40 binds the FERM-FA domains of FRM-3. a** Schematic representation of UNC-40 predicted domains: IG: Ig-like; FNIII: fibronectin type-III; TM: transmembrane region; P1, P2 and P3: conserved intracellular motifs. The putative truncated UNC-40 protein resulting from the *e1430* mutation is indicated. A hemagglutinin (HA) epitope tag was inserted to replace the last 33 amino acids containing the P3 region. **b** Transmitted-light images of wild-type, *unc-40* null (*e1430*) and *unc-40* ΔP3 (*kr457*) young adult animals crawling on plates (12 h after L4 stage). Scale bar = 100 μm. **c** Morphology of muscle cells assessed using a *Phim-4::yfp* reporter combined with a pan-neuronal *Prgef-1::dsRed* marker. Arrows, neurites. Stars, muscle arms. **d** Number of muscle arms per cell in the wild-type and *unc-40* mutants. Box plot as in Fig. 1g. One-way ANOVA followed by Tukey's multiple comparison test of each group compared with the wild type. ***$p < 0.001$, **$p < 0.01$. **e, f** Axonal morphology of the AVM touch neuron expressing a *Pmec-7::yfp* reporter. Scoring of guidance defects was repeated three times ($n = 35$ worms per genotype per individual experiment). Percentages of abnormal guidance events compared by one-way ANOVA followed by Tukey's multiple comparison test of each group. Box plot as in Fig. 1g. ***$p < 0.001$, **$p < 0.01$. **g** Actin organization visualized by COR-1-BFP expressed under the control of the *Phim-4* promoter in the outer rows of muscle quadrants. Stars: muscle arms; arrowheads: nerve cord. A SNB-1-GFP marker was expressed in GABAergic motoneurons to visualize synaptic boutons. Arrows: neuronal cell bodies. COR-1-BFP also labels sarcomeres (above the blue line, on the top of each picture). **h-j** Confocal detection of RFP-labeled GABA$_A$Rs expressed from the *unc-49::rfp* knock-in locus and *frm-3* FERM-FA-GFP expressed under the control of the muscle-specific promoter *Pmyo-3*. The overall fluorescence levels of GABA$_A$R (**i**) and of FERM + FA (**j**) were normalized to the wild type. Box plots are as in Fig. 1e. One-way ANOVA followed by Tukey's multiple comparison tests of each group compared with the wild type. ***$p < 0.001$. *$p < 0.05$. Scale bars = 10 μm except Fig. 5b where it is 100 μm.

postsynaptic components[44]. DCC participates to LTP in cortical neurons[1], and activity-dependent secretion of netrin potentiates excitatory glutamatergic transmission via the recruitment of GluA1-containing AMPA receptors[18]. In *C. elegans* UNC-40/DCC was initially described for its roles in cellular and axonal migration, but it was also shown to regulate presynaptic differentiation in response to local accumulation of UNC-6/netrin[24]. In all instances, the main underpinning mechanism was the reorganization of the actin cytoskeleton[11,45].

Here we propose a novel mechanism for UNC-40 involving the direct recruitment of FRM-3/FARP to synaptic regions. We showed that the P3 domain of UNC-40 binds the FERM domain of FRM-3/FARP in vitro. To validate the role of this interaction in vivo, we deleted the P3-coding region in the *unc-40* locus. Surprisingly, the ΔP3 mutants were much less sick than the null mutants, although P3 was shown to be required for dimerization of DCC intracellular domain, a key-event upon netrin binding to activate downstream signaling pathways[41]. In ΔP3, the developmental events involving an interaction between UNC-40 and UNC-5 might be preserved because heterodimerization is thought to require the P1 domain of DCC[12]. Moreover, actin regulators mostly require the P1 and P2 domains of UNC-40 rather than P3[37]. Accordingly, we observed less severe penetrance of guidance defects in ΔP3 as compared with *unc-40* null mutants. Actin bundles present in muscle arms and along the nerve cords of WT animals were partially maintained in ΔP3. By contrast, the loss of FRM-3 and GABA$_A$Rs was indistinguishable between ΔP3 and *unc-40* null mutants. These results do not rule out a role of the actin cytoskeleton for postsynaptic organization, yet these results point to an anchoring role of the UNC-40 protein in the recruitment of scaffolding molecules at the synapse.

How Punctin triggers the recruitment of FRM-3 at synapses through UNC-40 remains to be further explored. Punctin was demonstrated to bind and localize UNC-40 at the tip of the muscle arms during development. Together with UNC-6/Netrin, it activates the GEF UNC-73/trio and actin remodelers including the WAVE complex[23] to direct membrane extension, independently from FRM-3. In the adult, Punctin localizes UNC-40 at synapses. It might induce the formation of high order multimers of UNC-40, as suggested for DCC upon binding of netrin, and create an intracellular platform sufficient for FRM-3 binding. However, this interaction might also be regulated. For example, the FERM domain protein ezrin associates with DCC in a netrin-1–dependent manner, which requires FERM phosphorylation upon DCC-dependent Src and Rho kinase activation and DCC phosphorylation[46]. Finally, we observed that the synaptic recruitment of FRM-3 is not fully abolished in the *unc-40(0)* mutant, suggesting that other pathways can drive synaptic targeting and stabilization of FRM-3.

Our study has identified a critical role of FRM-3 in promoting GABA$_A$R clustering. FRM-3 was initially related to the band 4.1 (EPB4.1) protein[33], but the more recently identified isoform FRM-3B is the unambiguous ortholog of the mammalian FARP1 and FARP2. In neurons, the FARP proteins have been mostly implicated in the regulation of actin dynamics through GEF activity in response to guidance cues such as Sem6A/PlexA4[47] or Sema3A[48,49]. FARP1 directly interacts with the synaptogenic adhesion molecule SynCAM and regulates synapse number and dendritic spine morphology by activating Rac1 and regulating F-actin assembly[50]. In this study we demonstrated that the FERM-FA domain of FRM-3 is necessary and sufficient to rescue the synaptic defects of *frm-3(0)* mutants. Consistently, actin concentration is still detected in postsynaptic regions of *frm-3(0)* mutant, pointing to a specific function of the FERM-FA domain in the formation of a postsynaptic scaffold. This versatility might reflect an evolutionary diversification of FRM-3 functions, although a similar role of FARP proteins might be relevant at some synapses in mammals.

Interestingly, FERM-FA domains can oligomerize and bind protein and lipid partners. In *Drosophila*, the FERM protein Yurt, an orthologue of mammalian EPB41L5, controls epithelial cell polarity[38]. Oligomerization of Yurt is crucial for its function in epithelial cell polarity and is supported by the FERM-FA domains. In Yurt, the oligomerization interface relies on a hydrophobic FxW motif which is conserved in FRM-3. In addition, the crystal structure of the zebrafish FARP1 and FARP2 FERM domains identified a positively charged patch that supports phospholipid binding and targeting to the plasma membrane[51]. This KRKK motif is also present in FRM-3 (Supplementary Fig. 1a). Altogether, our results suggest a model in which FRM-3 might build up an oligomeric platform tethered to the plasma membrane, and recruit the adaptor protein LIN-2/CASK.

LIN-2 is the ortholog of CASK, a member of the membrane-associated guanylate kinase family. Most studies have focused mainly on presynaptic roles for CASK. Interestingly, CASK interacts with the synaptic adhesion molecule neurexin via its PDZ domain and supports local actin assembly through the FERM protein P4.1[52]. However, CASK is also detected in dendritic spines[53] and few studies report a role of CASK at the postsynaptic side. At *Drosophila* NMJs, knock-down of CASK in muscles induces smaller glutamate-gated currents and triggers specific loss of GluRIIA subunits of glutamate receptors[54]. CASK can also function as a co-activator of TBR1, a T-box transcription factor, and enhance the transcription of genes such as the N-methyl-D-aspartate (NMDA) receptor subunit GluN2B[55,56]. In the present study we propose that LIN-2 is recruited to synaptic regions by FRM-3 and interacts with both neuroligin via its PDZ domain and

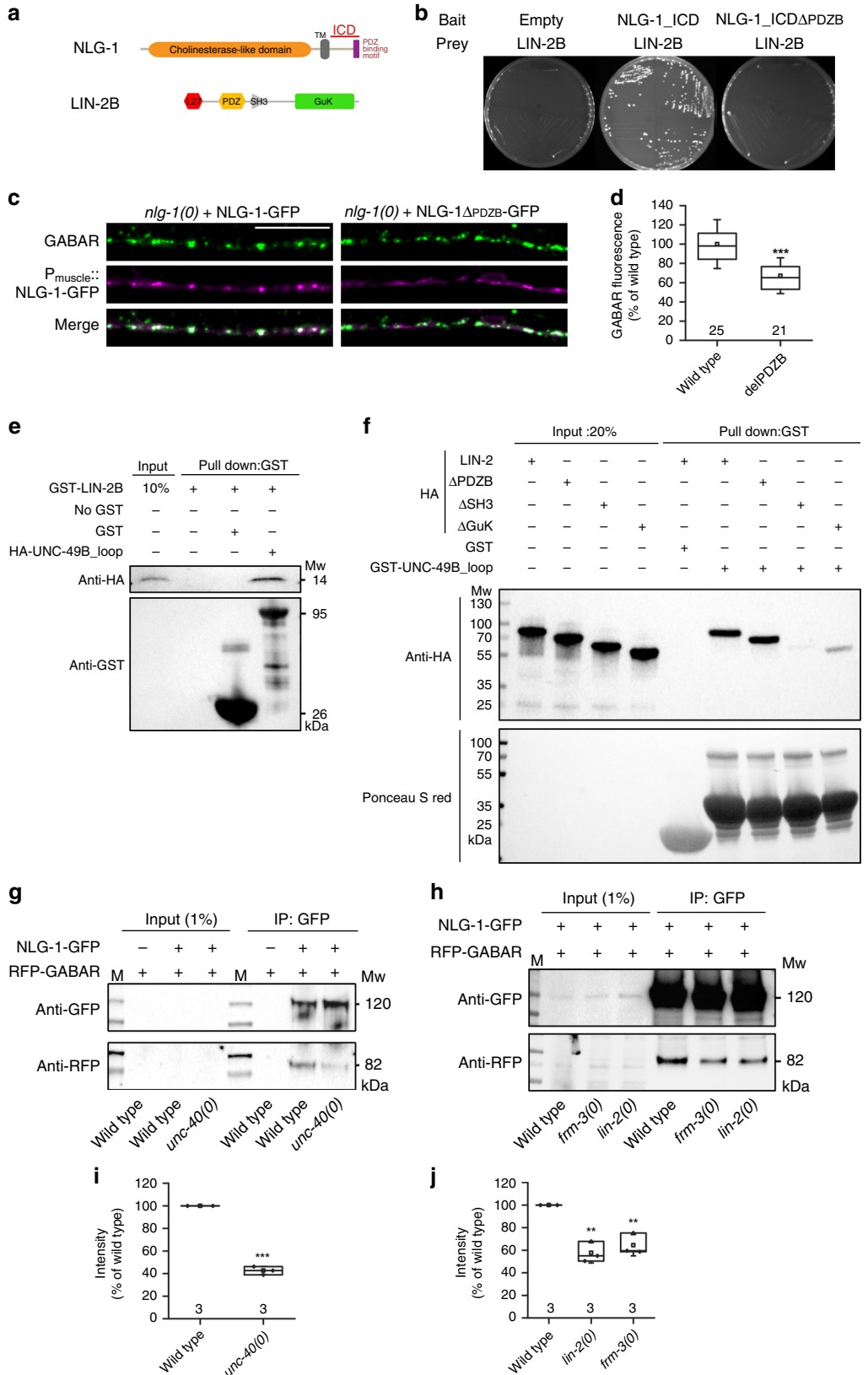

with the TM3-TM4 cytoplasmic loop of the GABA$_A$R via its SH3 and guanylate-kinase domains.

Our model differs from a previous study[33] that proposed the existence of two GABA$_A$R pools at the *C. elegans* NMJs, one synaptic pool depending on NLG-1/neuroligin, and a second synaptic pool depending on FRM-3/LIN-2. In this model, GABA$_A$Rs were proposed to interact with FRM-3 based on a yeast-2-hybrid experiment. Although we could not detect this

**Fig. 6 LIN-2 binds to the TM3-4 loop of UNC-49 and to NLG-1. a** Domain composition of NLG-1 and LIN-2B. **b** LIN-2B binds NLG-1 intracellular domain (NLG-1_ICD) in yeast two-hybrid assays. By contrast, LIN-2B does not bind NLG-1 after deletion of the PDZ binding motif (NLG-1_ICDΔPDZB). **c, d** Confocal detection of RFP-labeled GABA_ARs in *nlg-1(0)* null mutants expressing full length NLG-1-GFP or NLG-1_ΔPDZB-GFP in muscle. GABA_AR fluorescence levels were normalized to the wild type. Box plot as in Fig. 1e. Two-tailed Student's t test, \*\*\*p < 0.001. **e** Direct binding of GST-LIN-2B and HA-UNC-49B_TM3-4 intracellular loop was detected in GST pull down assay. The samples were analyzed by immunoblotting using an anti-HA antibody. The same membrane was stained with an anti-GST antibody to show GST expression. The experiment was repeated twice. **f** Pull down experiment using GST-tagged UNC-49B_loop and HA-tagged full-length and deleted versions of LIN-2B. Samples were analyzed by immunoblotting using an anti-HA antibody. The same membrane was stained by ponceau S red to show GST expression. **g, h** Co-immunoprecipitation from extracts of wild-type and *unc-40(0)*, *lin-2(0)* and *frm-3(0)* null mutants expressing RFP-UNC-49 and NLG-1-GFP in muscle. Worm lysates were precipitated with anti-GFP nanobody-coupled TrapA beads. Samples were analyzed by western blot using anti-RFP or GFP antibodies. **i, j** RFP band intensities were normalized to GFP bands. Data are presented as RFP/GFP ratios normalized to the wild type. Box plots are as in Fig. 1g. Two-tailed Student's *t* test \*\*\*p < 0.001, \*\*p < 0.01. The experiment was repeated three times. Scale bar = 10 μm.

interaction by in vitro pull-down or yeast-2-hybrid (unpublished data), this interaction might exist. Yet, genetic data point to a central role of LIN-2 because we observe the same dramatic loss of GABA_ARs in *lin-2(0)* and in *frm-3(0)* mutants while FRM-3 remains enriched at synapses in *lin-2(0)*. This is strong contrast with *nlg-1(0)* mutants where receptors are no longer clustered at GABA synapses but remain enriched along the cord, most probably by interacting with the FRM-3/LIN-2 complex. Removing LIN-2 in the presence or absence of NLG-1 causes the same loss of receptors, in contrast with previous studies which suggested that *lin-2* and *nlg-1* were synergistic. Few hypotheses might be proposed to reconcile these data. First, the previously used allele *lin-2(e1309)* might retain some expression of the LIN-2B isoform, which we showed to be sufficient for GABA_AR clustering. Second, GABA_AR localization and dynamics analysis was performed using transgenic expression of the UNC-49B subunit tagged in its TM3-TM4 loop, while we used CRISPR/Cas9 to modify the *unc-49* locus and fuse RFP at the shared N-terminal part of the UNC-49 subunits. Tagged-UNC-49B will form a homomeric pentamer containing five fluorescent proteins in its cytoplasmic region while in our N-terminally tagged knock-in it will likely form a heteromeric B/C heteropentamer as thought for the endogenous receptor[57]. Third, since LIN-2 interacts with the cytoplasmic loop of UNC-49, we wondered if the presence of a fluorescent protein could modify the interactions with the FRM-3/LIN-2 scaffold. We therefore introduced by CRISPR/Cas9 the wormScarlet protein into the UNC-49B loop. This receptor localizes at GABA synapses but we observed only a 50% decrease of fluorescence in *lin-2(0)* mutants as compared with more than 80% when using the extracellularly tagged reporter (Supplementary Fig. 8a–b). Yet the FERM-FA domains and the GABA_AR interact in both constructs (Supplementary Fig. 8c). These data suggest that the presence of a fluorescent protein in the TM3-TM4 loop of UNC-49B might weaken the contribution of LIN-2 at GABA synapses in vivo and suggest that parallel, yet uncharacterized, mechanisms might additionally support the recruitment of UNC-49 at neuroligin clusters. However, the dramatic loss of synaptic GABA-evoked currents in *frm-3(0)* and *lin-2(0)* mutants unambiguously demonstrates that the FRM-3/LIN-2 scaffold is critical for endogenous GABA_AR localization at the synapse.

The evolutionary strategy for GABA_AR clustering at *C. elegans* NMJs diverged from mammalian GABA synapses. In mammals, the self-oligomerizing protein gephyrin (GPN) is the core component of inhibitory synapses that binds the intracellular region of GABA_ARs to anchor them at the synapse[58,59]. Collybistin (CB), a Rho-guanine nucleotide exchange factor, binds GPN and GABA_ARs, which promotes the interaction of each protein with the intracellular region of Neuroligin-2. In addition, the transmembrane tetraspanins LHFPL3 and 4, also named GARLH (GABA_AR regulatory LHFPL), form a tripartite complex with GABA_ARs and Neuroligin 2 and control synaptic clustering, of

GABA_ARs[60,61]. In addition, GABA_AR GPN-independent clustering mechanisms exist because GABA_ARs still form functional synaptic clusters in some neurons of GPN knockout mice[62,63] (for complete discussion, see Tyagarajan and Fritschy, 2014)[59]. In extrasynaptic regions, the targeting of GABA_ARs containing the α4 or α5 subunit is ensured by Radixin, a FERM protein that bridges GABA_AR to the actin cytoskeleton[64–66]. GPN and CB are not present in the *C. elegans* genome, and the single *C. elegans* GARLH ortholog has not been involved in GABAergic transmission so far. However, it is remarkable to see that NLG-1, LIN-2/CASK and the intracellular domain of UNC-49 GABA_AR form a tripartite complex promoted by FRM-3/FARP upon UNC-40/DCC activation. Despite using different elementary components, this configuration is functionally reminiscent of the arrangement of postsynaptic inhibitory scaffolds. Interestingly, DCC was recently identified in a proteomic analysis of the synaptic cleft components at inhibitory synapses[67]. Whether DCC might control the content of GABA_ARs at inhibitory synapses in the mammalian brain remains to be investigated.

## Methods

**Strains and genetics**. All *C. elegans* strains were derived originally from the wild-type Bristol N2 strain. Worm cultures, genetic crosses, and other *C. elegans* manipulation were performed according to standard protocols[68]. All strains were maintained on nematode growth medium (NGM) agar plates with Escherichia coli OP50 as a food source at 20 °C. A complete list of strains used in this study can be found in Supplementary data 1.

Molecular information on *frm-3* (related to Fig. 1a). The *frm-3b* isoform is generated from alternative splicing in the last *frm-3a* exon. The *gk585* mutation is a deletion starting from Asp34, resulting in an early stop codon. The *kr319* allele is a G to A mutation in a splice donor site right after exon 7. These 2 mutations alter both isoforms.

Molecular information on *lin-2* (related to Fig. 1b). The *lin-2(e1309)* allele is a genomic rearrangement at the 5′ of *lin-2* locus. The *lin-2(n397)* and *lin-2(n105)* alleles are deletions affecting both *lin-2a* and *b* isoforms. *lin-2(kr357)* is a deletion spanning nucleotides 1218-1331 at the *lin-2a* locus (position 1 is the ATG of *lin-2a*), resulting in an early stop codon.

**Plasmids**. The constructs used and created in this study were described in Supplementary data 2. The full open reading frames of all constructs were verified by Sanger sequencing from GATC Company.

**Genome editing by MosTIC and CRISPR**. The final *tagRFP-T-unc-49(kr296)* or *pHluorin-unc-49(kr378)* knock-in were generated using a MosTIC based technique where the fluorescence tags were inserted between Asp26 and Ser27 residues.

To generate *kr392* allele, the wrmscarlet[69,70] sequence was inserted between Met392 and Glu393 amino acid in UNC-49 genome encoding TM3-4 loop using CRISPR method based on (Dickinson et al., 2015)[71]. CrRNA sequence targeting insertion site of UNC-49 loop was designed and synthesized from IDT (Integrated DNA Technologies Inc.). CrRNA and tracerRNA (trRNA) were mixed as 1:1 ratio to form a duplex. Cr/trRNA duplex was incubated at 95 °C for 5 minutes and then placed at room temperature for 5 min. Injection mix contains Cas9 nuclease (10 μg) (IDT Inc.), cr/trRNA duplex 3 μL (100 μM), pOR2.2 repair template 50 ng/μL, pCFJ90 [Pmyo-2::mCherry] co-injection marker 2.5 ng/μL, RNAase/DNAase free water up to 10 μL. N2 young-adult animals were grown under 25 °C after injection. Hygromycin was added to plates 48 h later at concentration of 0.2 μg/μL for selection of resistant worms. Initial knock-in animals with roller phenotype were isolated and heat-shocked 2 h under 34 °C water bath to remove hygromycin

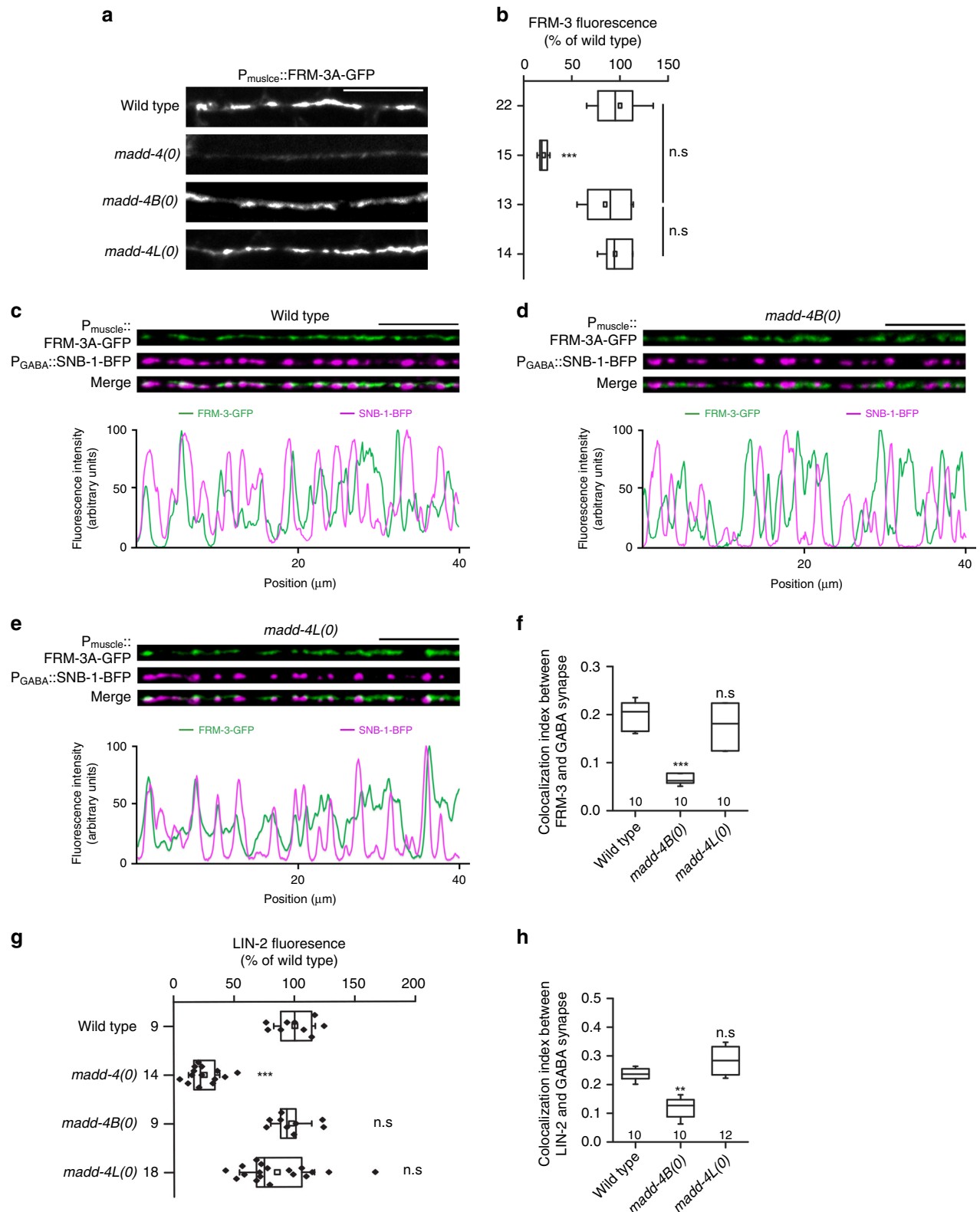

selection cassette from the genome. Non-roller worms were then isolated and the PCR products were then sent for sequencing. Candidates were outcrossed with N2 twice to remove unspecific background mutations and used for the further studies.

To generate *kr457* allele, the coding sequence of P3 region (Ala$^{1383}$-Lys$^{1415}$) within the intracellular domain of UNC-40 was replaced by HA (Human influenza hemagglutinin) epitope tag sequence. CrRNA was designed and synthesized from IDT (Integrated DNA Technologies Inc.). An ultramer DNA fragment containing

HA tag sequence, homology arms, NheI restriction enzyme site and a stop codon was produced by IDT Inc. Injection mix contains Cas9 nuclease (10 µg) (IDT Inc.), cr/trRNA duplex 3 µL (100 µM), ultramer repair template 1.25 µL (100 µM), pCFJ90 [*Pmyo-2::mCherry*] co-injection marker 2.5 ng/µL, RNAase/DNAase free water up to 10 µL. F1 progenies were checked by PCR and NheI restriction enzyme digestion (NheI site was introduced by ultramer repair template). Homozygotes were isolated and the PCR product were then sent for sequencing. Candidates were

**Fig. 7 MADD-4/Punctin controls the synaptic localization of the LIN-2-FRM-3 complex. a, b** Confocal detection of FRM-3A-GFP expressed in the muscle of wild-type animals and in mutants lacking the MADD-4 short isoform: *madd-4B(0)*; the long isoforms: *madd-4L(0)*; or all isoforms: *madd-4(0)*. FRM-3A-GFP fluorescence intensity was normalized to the wild type. Box plot as in Fig. 1e. One-way ANOVA followed by Turkey's multiple comparison tests. ***$p < 0.001$. n.s., not significant. **c, f** Confocal imaging of the dorsal cord of wild-type of animals expressing FRM-3A-GFP in muscle and SNB-1-BFP in GABA motoneurons to label presynaptic GABAergic boutons. The fluorescence profiles indicate FRM-3A-GFP and SNB-1-BFP fluorescence intensities along the nerve cord from the pictures above. **f** Mander's overlap correlation was used to calculate the percentage of FRM-3A-GFP signal that overlaps with the SNB-1-BFP signal. Box plot as in Fig. 2d. Kruskal–Wallis test followed by Dunn's post test. ***$p < 0.001$. n.s, not significant. **g, h** GFP-LIN-2A was expressed in the muscle of wild-type and mutant animals and SNB-1-BFP was used to label GABAergic boutons (images are shown in Fig S3). **g** GFP-LIN-2A fluorescence intensity was normalized to the wild type. Box plot as in Fig. 1g. One-way ANOVA followed by Turkey's multiple comparison tests. ***$p < 0.001$. n.s, not significant. **h** Mander's overlap correlation indicates the percentage of GFP-LIN-2A signal that overlaps with the SNB-1-BFP signal. Box plot as in Fig. 2d. Kruskal–Wallis test followed by Dunn's post test. **$p < 0.01$. n.s, not significant. Scale bars = 10 μm.

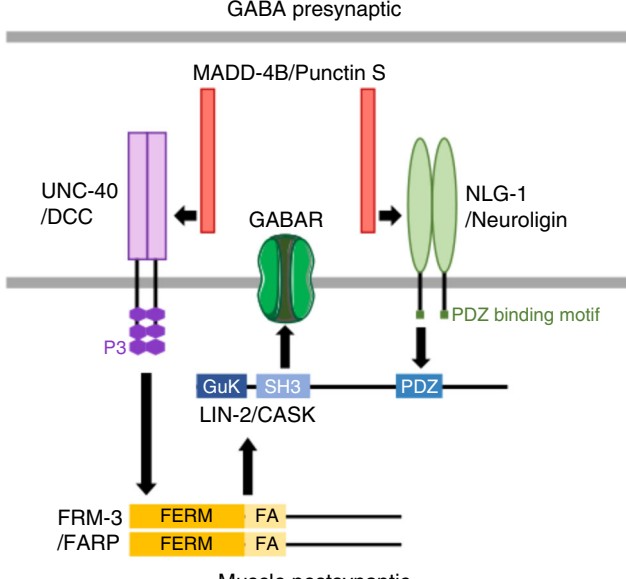

**Fig. 8 Working model for GABA$_A$R clustering at GABAergic neuromuscular junctions.** Activation of UNC-40 triggers the formation of an intracellular scaffold, which in turn promotes the interaction of GABA$_A$Rs with prepositioned Neuroligin/NLG-1 synaptic clusters. See Discussion for details.

outcrossed with N2 twice to remove unspecific background mutations and used for the further studies.

**Generation of single-copy insertion alleles**. The *krSi10* and *krSi28* allele encoding NLG-1-GFP or NLG-1_ΔPDZB-GFP (a deletion of the last 13 amino acid containing the PDZ binding motif) driven by *myo-3* body wall muscle specific promoter were generated by MosSCI[27,72]. The single-copy insertion alleles generated by the miniMos method[73] are listed in Supplementary Table. The fluorescent protein tagged genes are driven by *myo-3* body wall muscle promoter or by *him-4* distal muscle specific promoter. N2 worms were injected with 15 ng/μL of plasmid of interest containing the promoter and open reading frame, 50 ng/μL pCFJ601 (Mos1 transposase), 10 ng/μL pMA122 (negative selective marker *Phsp16.2::peel-1*), 2.5 ng/μL pCFJ90 (*Pmyo-2::mCherry*). Neomycin (G418) was added to plates 24 h after injection at 1.5 μg/μL final concentration. Candidate plates were heat-shocked for 2 h at 34 °C to remove extrachromosomal arrays. Worms with an insertion were isolated and homozygosed.

**Microscopy imaging and quantification**. Freely moving worms were observed on nematode growth media (NGM) plates using an AZ100 macroscope (Nikon) equipped with a Flash 4.0 CMOS camera (Hamamatsu Photonics).

For confocal imaging, young adult hermaphrodites (24 h post L4 larval stage) were used for imaging. Live worms were mounted on 2% agarose dry pads with 1% poly-lysine beads in M9 buffer. Fluorescence images were captured using an Andor spinning disk system (Oxford Instruments) installed on a Nikon-IX86 microscope (Olympus) equipped with a 60/1.2 oil immersion objective and an Evolve EMCCD camera. Images were acquired by IQ 3.4.1 software from Andor. Each animal was

imaged as a stack of optical sections (0.2 μm apart) containing 33–36 slices and projected along Z-axis.

Quantification of images was performed using ImageJ (v1.48 by NIH) with Fiji plugin add-ons. Statistic tests were performed by Prism6 from Graphpad and OriginLab software. The exact $P$ values for all the tests are provided in the source data file. For fluorescence intensity measurement, 30 μm (wide) × 5 μm (high) regions along ventral (between VD6 and VD7 neuron) or dorsal cord near the mid body were cropped and analyzed. Acquisition settings were the same across genotypes for quantitative analysis. For examination of each genetic background, each genotype of animals was imaged on at least 2 different days, and the data were pooled together. Data are presented as a percentage of the average fluorescence relative to that of the wild type. For co-localization quantification, images were captured along 40 μm in dorsal nerve cord anterior to the center of body. The plot file of either GFP-FRM-3/LIN-2/NLG-1 or SNB-1-BFP fluorescence signals were measured and processed by Mander's overlapping correlation analysis modified from previous studies[74,75]. The co-localization between FRM-3, LIN-2, GABAR and NLG-1 were analyzed by Pearson's coefficient correlation. Data are presented as min to max value for animals of each genotypes. For muscle arms growth quantification, number of arms from two body wall muscle cells in the quadrant connecting ventral nerve cord was counted. Average numbers of muscle arms per cell were compared between the wild type, *unc-40(e1430)*, *unc-40(kr457)* or *frm-3 (gk585)* mutants. For AVM neuron guidance quantification, pathfinding of AVM axon was characterized under spinning disk microscopy with a 60× optic lens. Ratio of abnormal/total number of animals counted from three independent days were compared between the wild type, *unc-40(e1430)* or *unc-40(kr457)* mutants. For the analysis of GABAR cluster size, the ROI of images were first thresholded by unique method provided by ImageJ. Then images were converted to 8-bit binary mask. The area of particles in each binary image was calculated. Average sizes were normalized to the mean of the wild type.

**Electrophysiology**. The strains used for electrophysiology were derived from ZX426 strain[76]. Microdissection of *C. elegans* was adapted from previous work:[77] the dissected region was always restrained to a small region (around 100 μm) before the vulva. Membrane currents were recorded in the whole-cell configuration using a MultiClamp 700B amplifier (Molecular Devices). Acquisition and command voltage were done using the Clampex 10 software driving an Axon Digidata 1550 (Molecular Devices). The bath solution contained (in mM) 150 NaCl, 5 KCl, 5 CaCl$_2$, 1 MgCl$_2$, 10 glucose, 15 HEPES, and sucrose to 320 mOsm/L (pH 7.2), and the pipette solution (in mM) 120 KCl, 4 NaCl, 5 EGTA, 10 TES, 4 MgATP and sucrose to 310 mOsm/L (pH 7.2). The slit worm preparations were exposed to 10 ms light stimulation performed with the pE2 system (CoolLED) at a wavelength of 460 nm drived by Clampex 10 software. The resistance of recording pipettes ranged between 3 and 4 MΩ. All chemicals were obtained from Sigma-Aldrich. All experiments were done at 20 °C. Data were analyzed with Clampfit 10 (Molecular Devices) and graphed with Origin software (OriginLab).

**Co-immunoprecipitation of *C. elegans* proteins**. Mixed stage worms of each genetic background were collected from 40 of 10 cm dishes with 0.1 mM NaCl. After washing three times at 4 °C, worm slurry was frozen in liquid nitrogen to form worm beads around 15 mL in volume. Five miililiters of worm beads was grinded in liquid nitrogen and thawed with 7.5 mL ice cold worm lysis buffer (WLB: 50 mM HEPES, 50 mM KCl, 100 mM NaCl, 1 mM EDTA, 2% Triton X-100, 2 mM PMSF and one tablet of cOmplete® Protease inhibitor cocktail (Roche) in 25 mL). Worm lysis was incubated at 4 °C for 2 h with rotation and then centrifuged at 15,000 × $g$ for 20 min at 4 °C. Supernatant was collected and diluted to a final concentration of 0.2% Triton-X100 by WLB. Pre-cleaned 100 μL Trap-A bead coupled with anti-GFP or RFP nanobody (Chromotek, gta-250 or rta-250) was added to the supernatant dilution and incubated at 4 °C overnight with gentle rotation. The beads were collected by centrifuge 1500 × $g$ at 4 °C and washed three times with washing buffer (50 mM HEPES, pH 7.7, 50 mM NaCl) containing protease inhibitor and PMSF without triton-X100. Beta-mercaptoethanol containing Laemmeli buffer was added as 1/4 volume to the washed beads. Precipitated proteins were eluted by heating at 95 °C for 10 min and centrifuge 1500 × $g$ at 4 °C

for 10 min. Forty microliters of supernatant was loaded to 4–20% gradient precast Tris-HEPES gel (ThermoScientifc, #0026244) per lane for each group. Proteins were transferred to Nitrocellulose membrane and incubated with mouse monoclonal anti-GFP (Roche #11814460001) or RFP (ThermoFisher Scientific, MA5-15257) antibody at 1:2000 dilution for 12 h at 4 °C. The membranes were then incubated with HRP conjugated secondary anti mouse antibody (Cell signaling 7076S) at 1:2000 dilution. Blot pictures were taken by Bio-rad ChemiDoc XRS + imaging system.

**GST-pulldown assay**. GST-pull down assay was modified from previously published protocol[78]. To obtain GST-fusion proteins, overnight cultures (5 mL) of *ArcticExpress E. coli* strain (Agilent, Santa Clara, CA, USA) transfected with pGEX-3X vectors were further cultured in 100 mL LB medium with 100 mg/ml ampicillin, grown for 2 h at 37 °C with shaking until OD600 reached 0.6–0.8. Protein expression was induced with 1 mM IPTG and shaking at 11 °C for 24 h. The culture was centrifuged at $5000 \times g$ for 10 min at 4 °C, the pellet was re-suspended in 10 mL ice-cold bacto-lysis buffer (50 mM HEPES pH 7.5, 400 mM NaCl, 1 mM DTT, 1 mM PMSF, 1 tablet of protease inhibitor cocktail/50 ml (Roche-Merck)). Cells were lysed on ice with 30 min of sonication. The lysed samples were then centrifuged at $15,000 \times g$ for 30 min at 4 °C. Supernatants were incubated with glutathione-Sepharose 4B beads (GE17-0756-01, Sigma) at 4 °C for 12 h. Coding sequences of FRM-3B, FERM + FA, FERM, LIN-2B or UNC-49B_loops were cloned into pcDNA3.1(+) vector with a HA epitope tag to the N-terminus for in vitro experiments. HA tagged proteins were transcribed and translated using TnT® Quick Coupled system (Promega, Madison, WI, USA), then incubated with GST-fusion coated beads for 12 h at 4 °C. The beads were then washed five times with 10 mM STE (10 mM Tris-Cl pH8.0, 150 mM NaCl, 1 mM EDTA, 0.1% Tween-20) buffer. Beads were boiled 10 min after the addition of an equal volume of 2×SDS sample buffer (Biorad, Hercules, CA, USA). Eluted proteins were separated on 4–20% gradient SDS/PAGE (Biorad, Hercules, CA, USA) and blotted to NC membrane. After blocking by 5% non-fatty milk, the blot was probed with anti-HA antibodies (#3724, Cell signaling tech.) at 1:1000 dilution, anti-GST (#2624, Cell signaling tech.) at 1:2000 dilution or stained immediately by ponceau S red.

**Immunohistochemical staining**. The N2, *frm-3(gk585)* or *lin-2(n397)* animals were grown and collected from NGM rich medium plates. A freeze/cracking step was performed before acetone/methanol fixation at −20 °C. Primary antibody rabbit anti-UNC-49 was diluted at 1:250; mouse anti-UNC-17 (VAChT) was diluted at 1:500[79]. Secondary antibodies Cy3-labeled goat anti-rabbit IgG was diluted at 1:500 (Molecular Probes, A10520), Alexa 488-labeled goat anti-mouse IgG ((Molecular Probes, A-11001) was diluted at 1:500.

**Yeast-two-hybrid assay**. Yeast-two-hybrid assays were performed by the Hybrigenics company. The coding sequence of the *Caenorhabditis elegans* NLG-1 fragment [aa.710-798] was cloned as a bait. The prey fragments for LIN-2 were: LIN-2B full length [aa. 1-620] (GenBank accession number X92565.1). The constructs were checked by sequencing the entire inserts. These assays are based on the *HIS3* reporter gene (growth assay without histidine). Controls and interactions were tested in the form of streaks of three independent yeast clones on DO-2 and DO-3 selective media. The DO-2 selective medium lacking tryptophan and leucine was used as a growth control and to verify the presence of the bait and prey plasmids. The DO-3 selective medium without tryptophan, leucine, and histidine selects for the interaction between bait and prey.

**Reporting summary**. Further information on research design is available in the Nature Research Reporting Summary linked to this article.

## Data availability

The full raw data that support the results of this study are available upon request. The source data underlying the quantification of Figs. 1e–h, 2d, g, j–l, 3c–f, i, j–k, n, 4a, 5d, 5f, 5i–j, 6d, i–j, b, 7f–h and Supplementary Figs. 1b–d, 1f, h, 2c, 3b, 4b–c, 4e, 6 and 8b are provided as a Source data file.

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

## Acknowledgements

We thank Haijun Tu for strain engineering, Josh Kaplan, Chun-Liang Pan and Guangshuo Ou for plasmids, the *Caenorhabditis* Genetic Center (which is funded by NIH Office of Research Infrastructure Programs, P40 OD010440) and Dr. Shohei Mitani for strains, Alexis Weinreb for image analysis, the CIQLE Imaging facility for support and access to equipment, the Hybrigenics company for fruitful interactions. X.Z. was supported by the ANR and ERC, M.G. was supported by the ERC, M.J. was supported by University Lyon 1. T.J. was supported by a fellowship from the China Scholarship Council, A.V. was supported by the Programme Avenir Lyon Saint-Etienne (PALSE), B.P.L. was supported by INSERM. This work was supported by the PALSE, the ANR grant GABAREL ANR-15-CE11-0016-01 and the European Research Council (ERC_Adg C.NAPSE #695295).

## Author contributions

Conceptualization, J.L.B., B.P.L., X.Z.; Methodology, X.Z., T.J., M.J.; Investigation, X.Z., M.G., M.J., A.V., T.J; Writing, X.Z., B.P.L. M.J. and J.L.B.

## Competing interests

The authors have no competing of interests.
