## [Peer Review File · Nature Communications]

Reviewers' Comments:

Reviewer #1:

Remarks to the Author:

This is exciting and important work that defines a new mechanism for post-synaptic organization, in this case GABA receptors in inhibitory synapses in the *C. elegans* neuromuscular system. The authors define very nicely a new signaling mechanism involving the UNC-40/DCC receptor, and a potential ligand MADD-4. This is an UNC-6/Netrin-independent role of UNC-40/DCC, and likely is independent of the well-known effects of DCC on the actin cytoskeleton. Indeed, the authors show that the P3 cytoplasmic domain of UNC-40 is required, which is not required in the other aspects of UNC-40 functions with actin. In their mechanism, MADD-4 causes DCC activation in post-synaptic regions, which recruits the FERM protein FRM-3. Their data suggest that FRM-3 might form a scaffold upon which LIN-2/CASK is recruited, which then recruits Neuroligin (NLG-1) and the UNC-49 GABA receptor, forming a post-synaptic structure in response to a signal from the motor neuron (MADD-4). While UNC-40/DCC and NLG-1 have been previously shown to be required for receptor clustering, the involvement of FRM-3 and LIN-2 is novel, as is the idea that LIN-2 might link UNC-40/DCC and NLG-1 to coordinately recruit GABA receptors.

The approach is a strong mix of genetics, which identified FRM-3 and LIN-2 in clustering, in vivo imaging of tagged molecules, immunofluorescence, and biochemistry to probe interactions in protein complexes and direct interactions.

The experiments are thorough and well-constructed and the appropriate controls and statistics are utilized.

The specificity of the phenotype to post-synaptic organization but not muscle arm formation is well-documented, as is the non-actin role of UNC-40 and the P3 domain in this process.

The experiments showing dependence of localization of one molecule on the presence of the others allow the authors to order the molecules in a mechanistic pathway.

Comments:

1) While not the main thrust of the manuscript, the description of UNC-6/Netrin signaling in the introduction could be updated to include results that cast doubt on the chemotactic gradient model of directed growth cone migration. This might be relevant to the work presented here, which is also likely short-range. Work in mice has shown that floorplate Netrin is largely dispensable for commissural axon guidance, and that rather ventricular zone netrin is important, possible in a close-range or contact-dependent manner (Domenici et al., 2017; Varadarajan et al., 2017; Varadarajan and Butler, 2017; Yamauchi et al., 2017). In *C. elegans*, the Statistically Oriented Asymmetric Localization (SOAL) model of growth toward UNC-6/Netrin suggests a cell polarity event involving UNC-40/DCC and UNC-5 that focuses UNC-40/DCC to the ventral side of the cell body where axon formation will occur (Limerick et al., 2018; Kulkarni et al., 2019). In axons that grow away from UNC-6/Netrin, UNC-5 first polarizes the growth cone such that protrusion occurs dorsally away from the UNC-6/Netrin source, and then regulates the extent of protrusion based on this initial polarity model (Norris et al. 2011; Gujar et al., 2018; Gujar et al., 2019). In this polarity/protrusion model, UNC-6/Netrin stimulates protrusion dorsally via UNC-40, and inhibits protrusion ventrally, via UNC-5, resulting in directed growth away from UNC-6/Netrin. In both SOAL and polarity/protrusion, the role of UNC-5 is to focus UNC-40 protrusive activity in the direction of growth. In each of these new models, chemotactic gradients need not be invoked, and in some cases might involve close-range or direct contact polarity events, rather than a gradient (vertebrate and polarity/protrusion). This sort of close-range interaction is likely also involved in the mechanism described here.

2) *unc-40(e1430)* likely has residual activity, as it is not as strongly *Unc* and *Dpy* as two other premature stop alleles *e271* and *n324*.

3) Are the conserved hydrophobic residues in the FERM domain required for FERM-FERM interaction?

4) The use of "epistatic" to describe the interaction of *frm-3* mutation relative to *nlg-1* in double mutants might not be appropriate. Epistasis is usually used to describe interactions where the phenotypes of two mutants are opposite (e.g. too much clustering versus not enough clustering). In the case of *nlg-1* and *frm-3*, they have the same phenotype, but *nlg-1* is weaker. This experiment does indeed suggest that they act in the same pathway, however.

Overall, this is an important piece of work that defines a new mechanism of post-synaptic GABA receptor clustering.

Erik A. Lundquist
The University of Kansas

Reviewer #2:

Remarks to the Author:

This manuscript by Zhou et al. is a thorough study describing the mechanistic underpinnings of GABAAR clustering in *C. elegans*. The conclusions of the paper are that UNC-40/DCC organizes an FRM-3 and Lin-2/CASK scaffold that is important for synaptic GABAAR clustering at GABAergic terminals. The authors fully characterize a direct interaction between FRM-3 and UNC-40 which links GABAARs to NLG-1 and is important for GABAAR synaptic content. This GABAAR clustering mechanism is also thought to be controlled by CePunctin/MADD-4. Together this work describes an interesting mechanism of postsynaptic clustering of GABAARs, which could be really informative for the GABAergic synaptic community, where understanding how GABAergic synapses are organized in the absence of the major scaffolding molecule, gephyrin, is of paramount importance. However, my main concern is that the findings are somewhat incremental, considering this lab's previous publication (Tu et al., 2015). In this paper, the group already identified UNC-40/DCC as a GABAAR clustering molecule and part of a signaling pathway responsible for the recruitment of GABAARs by NLG-1. In fact, the authors already acknowledge that their previous publication also shows that FRM-3 and Lin-2 impact GABAAR content at synapses (p. 6, para 2). Although the current manuscript nicely further delineates in detail how these factors work together to control GABAAR clustering at GABAergic synapses, I think that it lacks the conceptual advance and novelty required for publication in Nature Communications.

The paper is well-written, although there are a few grammatical errors. The data are on the whole convincing and the conclusions are well supported by the experiments.

Specific comments:

- The authors show effects on GABAAR synaptic clustering or 'content' in each figure, but do these deficits translate to inhibitory synaptic function? Electrophysiological analysis of mutants should be performed to test this.
- Figure 1E. It is mentioned in the manuscript that the GABAARs form 'extremely small puncta' in the mutants. Some measurement of GABAAR puncta and synaptic bouton area would be nice to accompany this statement.
- Figure 1J. A more dramatic LUT would aid the visualization of single channel images.
- Figure 2A-D. It is not clear what the conclusion is for these panels. In the example (2A) it does not look like very much of Lin-2 actually localizes to GABAAR clusters (which is shown by the linescan and very low Pearson's coefficient). From the image it seems that Lin-2 is everywhere and just at GABAAR clusters by chance (certainly not 'concentrated'). Why is there no imaging of FRM-3 and GABAARs? This seems important to support the idea that both of these proteins are acting at these synapses. This Figure leads to the question of whether Lin-2 and FRM-3 are always

together at GABAergic synapses (can you have one and not the other?) and whether you need both to maintain GABAAR content.

-Figure S2 shows the presence of Lin-2 and FRM-3 at cholinergic NMJs- the overlap here looks far more significant and should be quantified and commented on.

-A schematic diagram of the players involved and the model proposed would be very helpful.

-The title could be more specific.

Reviewer #3:

Remarks to the Author:

The molecular steps that orchestrate the assembly of synapses and control the abundance of synaptic components are – rather surprisingly- still incompletely understood. This study builds on Tu et al. (2015) from the Bessereau group that demonstrated that the ECM-like molecule Punctin/MADD4 regulates at inhibitory synapses of *C. elegans* neuromuscular junctions the clustering of GABA-A receptors via two anterograde pathways, with one involving the direct binding of MADD4/Punctin to the sole neuroligin in *C. elegans*, NLG-1, while the other involves MADD-4 binding to the Ig protein and Netrin receptor DCC/UNC-40. Zhou and colleagues here present an underlying mechanism. Their genetic screen for mutants with loss of synaptic GABA-A receptors at NMJs identifies hits in the FERM domain protein FRM-3/Farp and the PDZ protein LIN-2/CASK, which co-localize at NMJs. Mutations in either of them causes a striking 80% loss of endogenous GABA-A receptors from synapses, without altering postsynaptic muscle arm extensions. DCC is upstream of the synaptic localization of FRM-3 and in agreement, the authors show their direct binding and map the involved protein domains. The authors' identification and mutation of the FRM-3 binding site in DCC supports that its P3 domain appears only partially involved in early events such as axon guidance but is fully required for GABA-A receptor recruitment to NMJs. allowing to dissociate DCC roles in early vs synaptic development. Further, the FRM-3 FERM+FA domains are sufficient to rescue the loss of GABA-A receptors in *frm-3* mutants and directly bind to LIN-2. Searching for synaptic interaction partners of LIN-2, the authors show its interactions with NLG-1 and with the extracellular domain of GABA-A receptor subunits/Unc-49. Additional studies demonstrate that MADD-4B is required not only for the previously reported localization of GABA-A receptors to inhibitory NMJ synapses but also of the LIN-2/FRM-3 complex.

This study is well performed and reasoned, and the results define distinct, partially interdependent steps required for inhibitory synapse assembly at *C. elegans* NMJs. While the results are convincing, it will be important to further corroborate the roles of the LIN-2/FRM-3 complex in the assembly of inhibitory synapses to support the model as outlined in the major points below.

Major points.

1. Is the coIP of GABA-A receptors/UNC-49 and NLG-1 shown in Figure 6 abrogated in FRM-3 mutants? Similarly, does LIN-2 loss impair this coIP of UNC-49 and NLG-1? These experiments will test the model that the LIN-2/FRM-3 complex recruits GABA-A receptors to NLG-1 sites.

2. Can electrophysiological evidence be provided, similar to the recordings in Tu et al., to support that the LIN-2/FRM-3 complex acts as an inhibitory synaptic hub?

Minor points.

3. Can data be obtained whether Punctin and Netrin signaling serve together to assemble this inhibitory synaptic scaffold? This would provide evidence for a coincidence detection mechanism that controls this assembly. Even though the fact that Punctin/MADD4 binds to both DCC and NLG-1 may confound experimental approaches to test this, such data would further add to this work.

4. The information can be added that mammalian Neuroligin-1 ends in a PDZ type I motif and does not bind CASK, pointing to evolutionary differences in the use of the scaffolding mechanism described here.
5. The Discussion correctly refers to the distinction of FRM-3 and mammalian Farp1 in that the latter engages in postsynaptic actin assembly and in activating Rac1. The authors can state that this may reflect evolutionarily diversified uses of this molecule.
6. DCC roles in mammalian synapses that have been reported so far focus on the plasticity of excitatory transmission, including LTP. The Discussion can refer to this to explain that DCC/Farp signaling may act at least in mammals at excitatory as well as at inhibitory synapses.
7. The authors could provide a model, e.g. in the Supplement.
8. The labels of Figure 7 F,H can state that the graphs show colocalization indices for Frm-3 and Lin-2, respectively.
9. Statistical information, notably the n for each group, should be reported throughout the figure legends.

Reply to reviewers' comments:

Reviewer #1: Comments

We thank Dr. Lundquist for his very positive feedback and for his suggestions to improve the manuscript.

1) *While not the main thrust of the manuscript, the description of UNC-6/Netrin signaling in the introduction could be updated to include results that cast doubt on the chemotactic gradient model of directed growth cone migration. This might be relevant to the work presented here, which is also likely short-range. Work in mice has shown that floorplate Netrin is largely dispensable for commissural axon guidance, and that rather ventricular zone netrin is important, possible in a close-range or contact-dependent manner (Domenici et al., 2017; Varadarajan et al., 2017; Varadarajan and Butler, 2017; Yamauchi et al., 2017). In C. elegans, the Statistically Oriented Asymmetric Localization (SOAL) model of growth toward UNC-6/Netrin suggests a cell polarity event involving UNC-40/DCC and UNC-5 that focuses UNC-40/DCC to the ventral side of the cell body where axon formation will occur (Limerick et al., 2018; Kulkarni et al., 2019). In axons that grow away from UNC-6/Netrin, UNC-5 first polarizes the growth cone such that protrusion occurs dorsally away from the UNC-6/Netrin source, and then regulates the extent of protrusion based on this initial polarity model (Norris et al. 2011; Gujar et al., 2018; Gujar et al., 2019). In this polarity/protrusion model, UNC-6/Netrin stimulates protrusion dorsally via UNC-40, and inhibits protrusion ventrally, via UNC-5, resulting in directed growth away from UNC-6/Netrin. In both SOAL and polarity/protrusion, the role of UNC-5 is to focus UNC-40 protrusive activity in the direction of growth. In each of these new models, chemotactic gradients need not be invoked, and in some cases might involve close-range or direct contact polarity events, rather than a gradient (vertebrate and polarity/protrusion). This sort of close-range interaction is likely also involved in the mechanism described here.*

We agree that, although not central to this current study, introducing the long-range versus short range netrin signaling models is relevant since in our system UNC-40 is clearly involved in a haptotactic interaction. We now provide this additional information in the introduction with appropriate references:

" This long-range chemotactic gradient model has recently been revisited after analysis of axonal growth in mice where netrin expression was inactivated in specific subregions of the developing nervous system (Domenici et al., 2017, Varadarajan et al., 2017, Yamauchi et al., 2017). In these studies, the phenotypes were more consistent with a short-range haptotactic guidance model involving the interaction of growing axons with netrin present in the local environment. Detailed analysis of single axon outgrowth in *C. elegans*, together with in silico modeling, suggests that UNC-6/netrin biases the distribution of UNC-40/DCC at the membrane surface and stimulates an UNC-40-dependent protrusive activity towards growth direction (Gujar et al., 2018, Limerick et al., 2018)."

2) *unc-40(e1430) likely has residual activity, as it is not as strongly Unc and Dpy as two other premature stop alleles e271 and n324.*

To address the important point raised here, we analyzed GABA_ARs in *unc-40(e271)* and *unc-40(n324)* mutants and saw similar defects caused by the 3 mutant alleles. These data are presented in Fig. S3 and mentioned in the results section:

" Using *rfp-* and *pHluorin-unc-49* knock-in stains, we confirmed that the amount of either total or surface GABA_ARs was reduced in multiple loss-of-function *unc-40* allele mutants, including the reference allele *e1430* further referred to as *unc-40(0)* (Fig. 3A-B and Fig. S3)."

3) *Are the conserved hydrophobic residues in the FERM domain required for FERM-FERM interaction?*

This assumption is based on the work performed on Yurt, the *Drosophila* ortholog of FRM-3, which demonstrated that the conserved FxW motif is critical for FERM-FA dimerization (Gamblin et al., 2018). We have not experimentally tested this requirement in FRM-3 but we show the conservation in Fig. S1 and mention it in the text:

"A recent study reported that the FERM-FA domains of the *Drosophila* protein Yurt form oligomers, and that multimerization of Yurt supports its function in cell polarity (Gamblin et al., 2018). Oligomerization relies on several hydrophobic residues within the F3 lobe of the FERM domain, which are conserved in FRM-3 (Fig. S1)."

4) The use of "epistatic" to describe the interaction of frm-3 mutation relative to nlg-1 in double mutants might not be appropriate. Epistasis is usually used to describe interactions where the phenotypes of two mutants are opposite (e.g. too much clustering versus not enough clustering). In the case of nlg-1 and frm-3, they have the same phenotype, but nlg-1 is weaker. This experiment does indeed suggest that they act in the same pathway, however.

We agree that epistasis is usually used to describe interactions between mutant alleles causing phenotypes that are opposite. However, the mutant phenotypes of *nlg-1(0)* vs *frm-3(0)* and *lin-2(0)*—i.e. partial decrease of surface GABA_ARs with extrasynaptic relocalization vs drastic loss of GABA_ARs with remaining synaptic puncta— are different enough to unambiguously identify the "epistatic" versus "hypostatic" phenotype in the double mutants. We therefore believe that the classical notion of epistasis is relevant in this case to describe the genetic interactions between these different loci.

Overall, this is an important piece of work that defines a new mechanism of post-synaptic GABA receptor clustering.

Thank you for this comment.

Reviewer #2 (Remarks to the Author):

*This manuscript by Zhou et al. is a thorough study describing the mechanistic underpinnings of GABAAR clustering in *C. elegans*. The conclusions of the paper are that UNC-40/DCC organizes an FRM-3 and Lin-2/CASK scaffold that is important for synaptic GABAAR clustering at GABAergic terminals. The authors fully characterize a direct interaction between FRM-3 and UNC-40 which links GABAARs to NLG-1 and is important for GABAAR synaptic content. This GABAAR clustering mechanism is also thought to be controlled by CePunctin/MADD-4. Together this work describes an interesting mechanism of postsynaptic clustering of GABAARs, which could be really informative for the GABAergic synaptic community, where understanding how GABAergic synapses are organized in the absence of the major scaffolding molecule, gephyrin, is of paramount importance.*

We thank our reviewer for his/her comment.

However, my main concern is that the findings are somewhat incremental, considering this lab's previous publication (Tu et al., 2015). In this paper, the group already identified UNC-40/DCC as a GABAAR clustering molecule and part of a signaling pathway responsible for the recruitment of GABAARs by NLG-1. In fact, the authors already acknowledge that their previous publication also shows that FRM-3 and Lin-2 impact GABAAR content at synapses (p. 6, para 2). Although the current manuscript nicely further delineates in detail how these factors work together to control GABAAR clustering at GABAergic synapses, I think that it lacks the conceptual advance and novelty required for publication in Nature Communications.

We believe that our reviewer is referring to the work of Tong et al. (2015), which was not performed by our laboratory. Indeed, based on the phenotypes that we observed and the experiments that we performed, our conclusions on the role of FRM-3/FARP and LIN-2/CASK at GABA synapses are extremely different from previous models (see discussion, p. 15 last paragraph and following). Although this kind of judgement is very subjective, we do believe that our work brings conceptual advance and novelty.

The paper is well-written, although there are a few grammatical errors. The data are on the whole convincing and the conclusions are well supported by the experiments.

We thank our reviewer for these comments and we apologize for the grammatical errors. The entire text was carefully edited, and if this manuscript is accepted for publication, we are confident that the editing process will eliminate putative remaining mistakes.

Specific comments:

-The authors show effects on GABAAR synaptic clustering or 'content' in each figure, but do these deficits translate to inhibitory synaptic function? Electrophysiological analysis of mutants should be performed to test this.

We performed the requested experiments. To test the synaptic GABAergic responses, we recorded the evoked inhibitory currents stimulated by Channelrhodopsins (ChR2) expressed in GABA motoneurons (Nagel et al., 2003). We detected 75% decrease of inhibitory currents in *frm-3* mutants and 90% decrease in *lin-2* mutants as compared to wild-type animals. This physiological phenotype is dramatic and is highly consistent with the measured decrease of synaptic GABA_AR fluorescence. The electrophysiology recordings and statistics are now provided in Fig. 1F and 1G:

"To evaluate the functional consequence of *frm-3* and *lin-2* inactivation on GABAergic transmission, we recorded GABA-evoked currents in muscle upon optogenetic stimulation of GABA motoneurons. As compared to the wild type, we observed a 75% and 90% decrease of GABA-evoked currents in *frm-3* and *lin-2* null mutants, respectively (Fig. 1F-G)."

-Figure 1E. It is mentioned in the manuscript that the GABAARs form 'extremely small puncta' in the mutants. Some measurement of GABAAR puncta and synaptic bouton area would be nice to accompany this statement.

We performed these measurements. Results are provided in Fig. S1B-D.

-Figure 1J. A more dramatic LUT would aid the visualization of single channel images.

We changed the LUT to fire appearance.

-Figure 2A-D. It is not clear what the conclusion is for these panels. In the example (2A) it does not look like very much of Lin-2 actually localizes to GABAAR clusters (which is shown by the linescan and very low Pearson's coefficient). From the image it seems that Lin-2 is everywhere and just at GABAAR clusters by chance (certainly not 'concentrated'). Why is there no imaging of FRM-3 and GABAARs? This seems important to support the idea that both of these proteins are acting at these synapses. This Figure leads to the question of whether Lin-2 and FRM-3 are always together at GABAergic synapses (can you have one and not the other?) and whether you need both to maintain GABAAR content.

We understand the concern of our reviewer and we tried to clarify the presentation of these data. First, as pointed by our reviewer, LIN-2 indeed localizes to GABAergic AND cholinergic synapses. Because cholinergic synapses are twice more abundant than GABAergic synapses along the cords, the Pearson's coefficient is higher with cholinergic markers (now quantified in Fig. S2 as requested by our reviewer). Yet, LIN-2 is clearly present at GABAergic synapses, otherwise the Pearson's coefficient would be negative. To avoid a misleading interpretation of these data, we removed the word "concentrated" and rephrased the description of the results:

"FRM-3 and LIN-2 puncta localized at GABAergic synapses (Fig. 2A), but were also found in between GABAergic synapses (Fig. 2A and D). Indeed, we found that FRM-3B-GFP and GFP-LIN-2A were also present at cholinergic NMJs (Fig. S2A-C)."

Second, the colocalization between LIN-2 and FRM-3 is extremely high (Pearson's coefficient at 0.87, very close to 1). And when we perturb LIN-2/FRM-3 distribution such as in a *madd-4B(0)* background, they always relocalize together (Fig. 7 and Fig. S7). This is consistent with the biochemical interaction between LIN-2 and FRM-3 and the identical phenotypes of *lin-2(0)* and *frm-3(0)* single and double mutants. All the

results of our study point to a necessary interaction of LIN-2 and FRM-3 to mount a physical hub connecting UNC-49 and NLG-1 (see graphical summary in Fig. S8). We have not included an image of FRM-3 with GABA_ARs to limit the presentation of repetitive images in already plethoric figures.

-Figure S2 shows the presence of Lin-2 and FRM-3 at cholinergic NMJs- the overlap here looks far more significant and should be quantified and commented on.

As mentioned above, we quantified the colocalization of LIN-2/FRM-3 at cholinergic synapses. As anticipated by our reviewer, this localization underlies a functional role of LIN-2 and FRM-3 at cholinergic synapses, which we are currently analyzing, but this goes beyond the scope of this current manuscript.

-A schematic diagram of the players involved and the model proposed would be very helpful.

This is now provided in Figure S8.

-The title could be more specific.

The current title already reaches the maximal size allowed at *Nature Communications*. We asked our editor whether the title should be modified but did not receive any recommendation into that direction.

Reviewer #3 (Remarks to the Author):

The molecular steps that orchestrate the assembly of synapses and control the abundance of synaptic components are – rather surprisingly- still incompletely understood. This study builds on Tu et al. (2015) from the Bessereau group that demonstrated that the ECM-like molecule Punctin/MADD4 regulates at inhibitory synapses of *C. elegans* neuromuscular junctions the clustering of GABA-A receptors via two anterograde pathways, with one involving the direct binding of MADD4/Punctin to the sole neuroligin in *C. elegans*, NLG-1, while the other involves MADD-4 binding to the Ig protein and Netrin receptor DCC/UNC-40. Zhou and colleagues here present an underlying mechanism. Their genetic screen for mutants with loss of synaptic GABA-A receptors at NMJs identifies hits in the FERM domain protein FRM-3/Farp and the PDZ protein LIN-2/CASK, which co-localize at NMJs. Mutations in either of them causes a striking 80% loss of endogenous GABA-A receptors from synapses, without altering postsynaptic muscle arm extensions. DCC is upstream of the synaptic localization of FRM-3 and in agreement, the authors show their direct binding and map the involved protein domains. The authors' identification and mutation of the FRM-3 binding site in DCC supports that its P3 domain appears only partially involved in early events such as axon guidance but is fully required for GABA-A receptor recruitment to NMJs. allowing to dissociate DCC roles in early vs synaptic development. Further, the FRM-3 FERM+FA domains are sufficient to rescue the loss of GABA-A receptors in *frm-3* mutants and directly bind to LIN-2. Searching for synaptic interaction partners of LIN-2, the authors show its interactions with NLG-1 and with the extracellular domain of GABA-A receptor subunits/Unc-49. Additional studies demonstrate that MADD-4B is required not only for the previously reported localization of GABA-A receptors to inhibitory NMJ synapses but also of the LIN-2/FRM-3 complex.

This study is well performed and reasoned, and the results define distinct, partially interdependent steps required for inhibitory synapse assembly at *C. elegans* NMJs. While the results are convincing, it will be important to further corroborate the roles of the LIN-2/FRM-3 complex in the assembly of inhibitory synapses to support the model as outlined in the major points below.

Major points.

1. Is the colP of GABA-A receptors/UNC-49 and NLG-1 shown in Figure 6 abrogated in FRM-3 mutants?

Similarly, does LIN-2 loss impair this colP of UNC-49 and NLG-1? These experiments will test the model that the LIN-2/FRM-3 complex recruits GABA-A receptors to NLG-1 sites.

The point raised by our reviewer is indeed a clear prediction of our model. We performed the requested colPs in *lin-2(0)* and *frm-3(0)* mutant backgrounds and observed a decreased efficiency of retrieving GABA_ARs upon IP of NLG-1, in agreement with our proposed model. These results are shown in Fig. 6I-K and are mentioned in the result section:

"Second, the interaction between NLG-1 and UNC-49 should depend on FRM-3 and LIN-2 if they provide a hub to link NLG-1 and UNC-49. Accordingly, the colP efficiency between NLG-1 and UNC-49 was reduced in *frm-3(0)* or *lin-2(0)* mutant animals (Fig. 6I and K)."

2. Can electrophysiological evidence be provided, similar to the recordings in Tu et al., to support that the LIN-2/FRM-3 complex acts as an inhibitory synaptic hub?

We performed the requested experiments. To test the synaptic GABAergic responses, we recorded the evoked inhibitory currents stimulated by Channelrhodopsins (ChR2) expressed in GABA motoneurons (Nagel et al., 2003). We detected 75% decrease of inhibitory currents in *frm-3(0)* mutants and 90% decrease in *lin-2(0)* mutants as compared to wild-type animals. This physiological phenotype is dramatic and is highly consistent with the measured decrease of synaptic GABA_AR fluorescence. The electrophysiology recording and statistics are now provided in Fig. 1F and 1G:

"To evaluate the functional consequence of *frm-3* and *lin-2* inactivation on GABAergic transmission, we recorded GABA-evoked currents in muscle upon optogenetic stimulation of GABA motoneurons. As compared to the wild type, we observed a 75% and 90% decrease of GABA-evoked currents in *frm-3* and *lin-2* null mutants, respectively (Fig. 1F-G)."

Minor points

3. Can data be obtained whether Punctin and Netrin signaling serve together to assemble this inhibitory synaptic scaffold? This would provide evidence for a coincidence detection mechanism that controls this assembly. Even though the fact that Punctin/MADD4 binds to both DCC and NLG-1 may confound experimental approaches to test this, such data would further add to this work.

We previously reported that disruption of *unc-6/netrin* causes a 37% decrease of GABA_AR. However, *madd-4; unc-6* double mutants do not exhibit more defects than *madd-4* single mutants (Tu et al., 2015, Fig 6A). This is likely explained by the fact that UNC-40 receptors no longer localize at the synapse in *madd-4* mutants. Although we cannot rule out the appealing hypothesis of a coincidence mechanism between netrin and punctin, we cannot really test it because Punctin dominates UNC-40/DCC-dependent signaling at adult synapses.

4. The information can be added that mammalian Neuroligin-1 ends in a PDZ type I motif and does not bind CASK, pointing to evolutionary differences in the use of the scaffolding mechanism described here.

This information was added in the text (p.11):

"Unlike mammalian neuroligins that end with a type I PDZ motif not predicted to interact with CASK, NLG-1 ends with a type II PDZ-binding domain, characterized by a hydrophobic residue at position -2 with respect to the C-terminus (Fig. 6A)."

5. The Discussion correctly refers to the distinction of FRM-3 and mammalian Farp1 in that the latter engages in postsynaptic actin assembly and in activating Rac1. The authors can state that this may reflect evolutionarily diversified uses of this molecule.

This is now mentioned in the discussion (p.14):

"Consistently, actin concentration is still detected in postsynaptic regions of *frm-3(0)* mutant, pointing to a specific function of the FERM-FA domain in the formation of a postsynaptic scaffold. This versatility might reflect an evolutionary diversification of FRM-3 functions, although a similar role of FARP proteins might be relevant at some synapses in mammals."

6. DCC roles in mammalian synapses that have been reported so far focus on the plasticity of excitatory transmission, including LTP. The Discussion can refer to this to explain that DCC/Farp signaling may act at least in mammals at excitatory as well as at inhibitory synapses.

This is an extremely interesting point. However, very few elements, so far, support the presence of DCC/Farp signaling at mammalian synapses. According to Cheadle and Biederer (2012), it seems that Farp1 is enriched at dendritic spine area and activates Rac1 to promote local actin assembly. From a more recent proteomic work analyzing molecules localized in the synaptic cleft (Loh et al., 2016), DCC was found at both glutamatergic and GABAergic synapses. These studies are cited in our manuscript. At this stage, we feel that we miss information on DCC/Farp signaling to speculate farther on the implication of this pathway at mammalian synapses.

7. The authors could provide a model, e.g. in the Supplement.

This is now provided in Figure S8.

8. The labels of Figure 7 F,H can state that the graphs show colocalization indices for Frm-3 and Lin-2, respectively.

The pictures have been modified accordingly.

9. Statistical information, notably the n for each group, should be reported throughout the figure legends.

We carefully went throughout the manuscript. Statistical information, notably the n for each group, is provided for each experiment, either directly in the bar graphs or in the legends.

Reviewers' Comments:

Reviewer #1:

Remarks to the Author:

The authors have satisfactorily addressed my comments.

Erik Lundquist

Reviewer #2:

Remarks to the Author:

I am happy with the changes that the authors have made to the manuscript, and I think it is now ready for publication.

Reviewer #3:

Remarks to the Author:

The authors have fully addressed my points through well designed and carefully performed new experiments. In particular, the reduced coIP of the NLG-1/UNC-49 complex in mutants of Farp/FRM-3 and LIN-2 is a very strong molecular validation of their model. The striking decrease in inhibitory currents in frm-3 and lin-2 mutants provides additional, compelling support. This work offers novel and important insights into how extracellular organizers, adhesion molecules, receptors, and scaffold proteins can put the assembly of inhibitory postsynaptic sites under local control. Thomas Biederer